# Faster Matchings via Learned Duals

**Michael Dinitz**
Johns Hopkins University
mdinitz@cs.jhu.edu

**Sungjin Im**
UC Merced
sim3@ucmerced.edu

**Thomas Lavastida**
Carnegie Mellon University
tlavasti@andrew.cmu.edu

**Benjamin Moseley**
Carnegie Mellon University
moseleyb@andrew.cmu.edu

**Sergei Vassilvitskii**
Google
sergeiv@google.com

## Abstract

A recent line of research investigates how algorithms can be augmented with machine-learned predictions to overcome worst case lower bounds. This area has revealed interesting algorithmic insights into problems, with particular success in the design of competitive online algorithms. However, the question of improving algorithm running times with predictions has largely been unexplored.

We take a first step in this direction by combining the idea of machine-learned predictions with the idea of "warm-starting" primal-dual algorithms. We consider one of the most important primitives in combinatorial optimization: weighted bipartite matching and its generalization to $b$-matching. We identify three key challenges when using learned dual variables in a primal-dual algorithm. First, predicted duals may be infeasible, so we give an algorithm that efficiently maps predicted infeasible duals to nearby feasible solutions. Second, once the duals are feasible, they may not be optimal, so we show that they can be used to quickly find an optimal solution. Finally, such predictions are useful only if they can be learned, so we show that the problem of learning duals for matching has low sample complexity. We validate our theoretical findings through experiments on both real and synthetic data. As a result we give a rigorous, practical, and empirically effective method to compute bipartite matchings.

## 1 Introduction

Classical algorithm analysis considers worst case performance of algorithms, capturing running times, approximation and competitive ratios, space complexities, and other notions of performance. Recently there has been a renewed interest in finding formal ways to go beyond worst case analysis [41], to better understand performance of algorithms observed in practice, and develop new methods tailored to typical inputs observed.

An emerging line of research dovetails this with progress in machine learning, and asks how algorithms can be augmented with machine-learned predictors to circumvent worst case lower bounds when the predictions are good, and approximately match them otherwise (see Mitzenmacher and Vassilvitskii [36] for a survey). Naturally, a rich area of applications of this paradigm has been in online algorithms, where the additional information revealed by the predictions reduces the uncertainty about the future and can lead to better choices, and thus better competitive ratios. For instance, see the work by Lykouris and Vassilvitskii [33], Rohatgi [40], Jiang et al. [29] on caching; Antoniadis et al. [3], Dütting et al. [19] on the classic secretary problem; Purohit et al. [39], Lattanzi et al. [32] on scheduling; Purohit et al. [39], Anand et al. [2] on ski rental; and Bamas et al. [8] on set cover.

35th Conference on Neural Information Processing Systems (NeurIPS 2021).

However, the power of predictions is not limited to improving online algorithms. Indeed, the aim of the empirical paper that jump-started this area by Kraska et al. [31] was to improve running times for basic indexing problems. The main goal and contribution of this work is to show that at least in one important setting (weighted bipartite matching), we can give formal justification for using machine learned predictions to improve running times: there are predictions which can provably be learned, and if these predictions are "good" then we have running times that outperform standard methods both in theory and empirically.

How can predictions help with running time? One intuitive approach, which has been used extensively in practice, is through the use of "warm-start" heuristics [46, 23, 22, 37], where instead of starting with a blank slate, the algorithm begins with some starting state (which we call a warm-start "solution" or "seed") which hopefully allows for faster completion. While it is a common technique, there is a dearth of analysis understanding what constitutes a good warm-start, when such initializations are helpful, and how they can best be leveraged.

Thus we have a natural goal: put warm-start heuristics on firm theoretical footing by interpreting the warm-start solution as learned predictions. In this set up we are given a number of instances of the problem (the training set), and we can use them to compute a warm-start solution that will (hopefully) allow us to more quickly compute the optimal solution on future, test-time, instances. There are three challenges that we must address:

(i) **Feasibility.** The learned prediction (warm-start solution) might not even be *feasible* for the specific instance we care about! For example, the learned solution may be matching an edge that does not exist in the graph at testing time.

(ii) **Optimization.** If the warm-start solution is feasible and near-optimal then we want the algorithm to take advantage of it. In other words, we would like our running time to be a function of the quality of the learned solution.

(iii) **Learnability.** It is easy to design predictions that are enormously helpful but which cannot actually be learned (e.g., the "prediction" is the optimal solution). We need to ensure that a typical solution learned from a few instances of the problem generalizes well to new examples, and thus offers potential speedups.

If we can overcome these three challenges, we will have an *end-to-end* framework for speeding up algorithms via learned predictions: use the solution to challenge (iii) to learn the predictions from historical data, use the solution to challenge (i) to quickly turn the prediction into something feasible for the particular problem instance while preserving near-optimality, and then use this as a warm-start seed in the solution to challenge (ii).

## 1.1   Our Contributions

We focus on one of the fundamental primitives of combinatorial optimization: computing bipartite matchings. For the bipartite minimum-weight perfect matching (MWPM) problem, as well as its extension to $b$-matching, we show that the above three challenges can be solved.

A key conceptual question is finding a specification of the seed, and an algorithm to use it that satisfies the desiderata above. We have discussed warm-start "solutions", so it is tempting to think that a good seed is a partial solution: a set of matched edges that can then be expanded to a optimal matching. After all, this is the structure we maintain in most classical matching algorithms. Moreover, any such solution is feasible (one can simply set non-existing edges to have very high weight), eschewing the need for the feasibility step. At the same time, as has been observed previously in the context of online matchings [13, 45], this *primal* solution is brittle, and a minor modification in the instance (e.g. an addition of a single edge) can completely change the set of optimal edges.

Instead, following the work of [13, 45], we look at the *dual* problem; that is, the dual to the natural linear program. We quantify the "quality" of a prediction $\hat{y}$ by its $\ell_1$-distance from the true optimal dual $y^*$, i.e., by $\|\hat{y} - y^*\|_1$. The smaller quantities correspond to better predictions. Since the dual is a packing problem we must contend with feasibility: we give a simple linear time algorithm that converts the prediction $\hat{y}$ into a feasible dual while increasing the $\ell_1$ distance by a factor of at most 3.

Next, we run the Hungarian method starting with the resulting feasible dual. Here, we show that the running time is in proportional to the $\ell_1$ distance of the feasible dual to the optimal dual (Theorem 5). Finally, we show via a pseudo-dimension argument that not many samples are needed before the

empirically optimal seed is a good approximation of the true optimum (Theorem 6), and that this empirical optimum can be computed efficiently. For the learning argument, we assume that matching instances are drawn from a fixed but unknown distribution $\mathcal{D}$.

Putting it all together gives us our main result.

**Theorem 1** (Informal). *There are three algorithms (feasibility, optimization, learning) with the following guarantees.*

- *Given a (possibly infeasible) dual $\hat{y}$ from the learning algorithm, there exists an $O(m + n)$ time algorithm that takes a problem instance $c$, and outputs a feasible dual $\hat{y}'(c)$ such that $\|\hat{y}'(c) - y^*(c)\|_1 \leq 3\|\hat{y} - y^*(c)\|_1$.*
- *The optimization algorithm takes as input feasible dual $\hat{y}'(c)$ and outputs a minimum weight perfect matching, and runs in time $\tilde{O}(m\sqrt{n} \cdot \min\{\|\hat{y}'(c) - y^*(c)\|_1, \sqrt{n}\})$.*
- *After $\tilde{O}(C^2 n^3)$ samples from an unknown distribution $\mathcal{D}$ over problem instances, the learning algorithm produces duals $\hat{y}$ so that $\mathbb{E}_{c \sim \mathcal{D}}[\|\hat{y} - y^*(c)\|_1]$ is approximately minimum among all possible choices of $\hat{y}$, where $C$ is the maximum edge cost and $y^*(c)$ is an optimal dual for instance $c$.*

*Combining these gives a single algorithm that, with access to $\tilde{O}(C^2 n^3)$ problem instance samples from $\mathcal{D}$, has expected running time on future instances from $\mathcal{D}$ of only $\tilde{O}(m\sqrt{n} \min\{\alpha, \sqrt{n}\})$, where $\alpha = \min_y \mathbb{E}_{c \sim \mathcal{D}}[\|y - y^*(c)\|_1]$.*

We emphasize that the Hungarian method with $\tilde{O}(mn)$ running time is the standard algorithm in practice. Although there are other theoretically faster exact algorithms for bipartite minimum-weight perfect matching [38, 21, 20, 18] that run in $O(m\sqrt{n}\log(nC))$, they are relatively complex (using various scaling techniques). Very recent breakthroughs give algorithms of run time $\tilde{O}((m + n^{1.5})\log^2(C))$ for the minimum-weight perfect matching problem and several interesting extensions [43, 44]. However, the algorithms are highly complicated and their practical performance is yet to be demonstrated. In fact, we could not find any implementation of the above algorithms, except for the Hungarian method, of which multiple implementations are readily available.

Note that our result shows that we can speed up the Hungarian method as long as the $\ell_1$-norm error of the learned dual, i.e., $\|\hat{y} - y^*(c)\|_1$ is $o(\sqrt{n})$. Further, as the projection step that converts the learned dual into a feasible dual takes only linear time, the overhead of our method is essentially negligible. Therefore, even if the prediction is of poor quality, our method has worst-case running time that is *never* worse than that of the Hungarian algorithm. Even our learning algorithm is simple, consisting of a straightforward empirical risk minimization algorithm (the analysis is more complex and involves bounding the "pseudo-dimension" of the loss functions).

We validate our theoretical results via experiments. For each dataset we first feed a small number of samples (fewer than our theoretical bounds) to our learning algorithm. We then compare the running time of our algorithm to that of the classical Hungarian algorithm on new instances.

Details of these experiments can be found in Section 4. At a high level they show that our algorithm is *significantly* faster in practice. Further, our experiment shows only very few samples are needed to achieve a notable speed-up. This confirms the power of our approach, giving a theoretically rigorous yet also practical method for warm-start primal-dual algorithms.

## 1.2 Related Work

**Matchings and $b$-Matchings:** Bipartite matchings are one of the most well studied problems in combinatorial optimization, with a long history of algorithmic improvements. We refer the interested reader to Duan and Pettie [17] for an overview. We highlight some particular results here. If edges have no weights and thus the goal is to find the maximum cardinality matching (see Section 2 for a formal definition), the fastest running time had long been $O(m\sqrt{n})$ [14, 30, 27] until the recent breakthrough with $\tilde{O}(m + n^{1.5})$ running time was discovered [43]. We are interested in the weighted versions of these problems and when all edge weights are integral. Let $C$ be the maximum edge weight, $n$ be the number of vertices, and $m$ the number of edges. For finding exact solutions to the minimum weight perfect matching problem, the scaling technique leads to a running time of $O(m\sqrt{n}\log(C))$ [38, 21, 20, 18].

Large scale bipartite matchings have been studied extensively in the online setting, as they represent the basic problem in ad allocations [34]. While the ad allocation is inherently online, most of the methods precompute a dual based solution based on a sample of the input [13, 45], and then argue that this solution is approximately optimal on the full instance. In contrast, we strive to compute the *exactly optimal* solution, but use previous instances to improve the running time of the approach.

**Algorithms with Predictions:** Kraska et al. [31] showed how to use machine learned predictions to improve heavily optimized indexing algorithms. The original paper was purely empirical and came with no rigorous guarantees; recently there has been a flurry of work putting such approaches on a strong theoretical foundation, evaluating the benefit of augmenting classical algorithms with machine learned predictions, see [36] for a survey. Online and streaming algorithms in particular have seen significant successes, as predictions reveal information about the future and can help guide the algorithms' choices. This has led to the design of new methods for caching [33, 40, 29], scheduling [39, 32], frequency counting [12, 28, 1], and membership testing [35, 42] that can break through worst-case lower bounds when the predictions are of sufficiently high quality.

Most of the above work abstracts the predictions as access to an error-prone oracle and asks how to best use predictions: getting performance gains when the predictions are good, but limiting the losses when they are not. A related emergent area is that of data driven algorithm design [25, 7, 5, 4, 6, 11]. Here, the objective is to "learn" a good algorithm for a particular family of inputs. The goal is not typically to tie the performance of the algorithm to the quality of the prediction, but rather to show that the prediction makes sense; that is only a small number of problem samples are needed in order to ensure the learned algorithm generalizes to new data points.

## 1.3 Roadmap

We present our theoretical results on min-cost perfect bipartite matching in Section 3. The experiments are presented in Section 4. The extension to $b$-matching, as well as all missing proofs, can be found in the full version of this paper as supplementary materials.

## 2 Preliminaries

**Notation:** Let $G = (V, E)$ be an undirected graph. When $G$ is bipartite we will use $L$ and $R$ to refer to the two sides of the bipartition. We will let $N(i) := \{e \in E \mid i \in e\}$ be the set of edges adjacent to vertex $i$. For a set $S \subseteq V$, let $\Gamma(S)$ be the vertex neighborhood of $S$. For a vector $y \in \mathbb{R}^n$, we let $\|y\|_1 = \sum_i |y_i|$ be its $\ell_1$-norm. Let $\langle x, y \rangle$ be the standard inner product on $\mathbb{R}^n$.

**Linear Programming and Complementary Slackness:** Here we recall optimality conditions for linear programming that are used to ensure the correctness of some algorithms we present. Consider the primal-dual pair of linear programs below.

$$\min c^\top x; \quad Ax = b; \quad x \geq 0 \quad (P) \qquad \max b^\top y; \quad A^\top y \leq c \quad (D)$$

A pair of solutions $x, y$ for $(P)$ and $(D)$, respectively, satisfy complementary slackness if $x^\top (c - A^\top y) = 0$. The following lemma is well-known.

**Lemma 2.** *Let $x$ be a feasible solution for $(P)$ and $y$ be a feasible solution for $(D)$. If the pair $x, y$ satisfies complementary slackness, then $x$ and $y$ are optimal solutions for their respective problems.*

**Maximum Cardinality Matching:** Let $G = (V, E)$ be a bipartite graph on $n$ vertices and $m$ edges. A matching $M \subseteq E$ is a collection of non-intersecting edges. The Hopcroft-Karp algorithm for finding a matching maximizing $|M|$ runs in time $O(\sqrt{n} \cdot m)$ [26], which is still state-of-the-art for general bipartite graphs. For moderately dense graphs, a recent result by van den Brand et al. [43] gives a better running time of $\tilde{O}(m + n^{1.5})$ (where $\tilde{O}$ hides polylogarithmic factors).

**Minimum Weight Perfect Matching (MWPM):** Again, let $G = (V, E)$ be a bipartite graph on $n$ vertices and $m$ with costs $c \in \mathbb{Z}_+^E$ on the edges, and let $C$ be the maximum cost. A matching $M$ is *perfect* if every vertex is matched by $M$. The objective of this problem is to find a perfect matching $M$ minimizing the cost $c(M) := \sum_{e \in M} c_e$.

When looking for optimal solutions we can assume that $G$ is a complete graph by adding all possible edges not in $E$ with weight $Cn^2$. It is easy to see that any $o(n)$ approximate solution would not use any of these edges.

## 3 Faster Min-Weight Perfect Matching

In this section we describe how predictions can be used to speed up the bipartite Minimum Weight Perfect Matching (MWPM) problem.

The MWPM problem can be modeled by the following linear program and its dual – the primal-dual view will be very useful for our algorithm and analysis. We will sometimes refer to a set of dual variables $y$ as dual *prices*. Both LPs are well-known to be integral, implying that there always exist integral optimal solutions.

Suppose we are given a prediction $\hat{y}$ of a dual solution. If $\hat{y}$ is feasible, then by complementary slackness we can check if $\hat{y}$ represents an optimal dual solution by running a maximum cardinality matching algorithm on the graph $G' = (V, E')$, where $E' = \{e = ij \in E \mid \hat{y}_i + \hat{y}_j = c_{ij}\}$ is the set of tight edges. If this matching is perfect, then its incidence vector $x$ satisfies complementary slackness with $\hat{y}$ and thus represents an optimal solution by Lemma 2.

$$\min \quad \sum_{e \in E} c_e x_e \qquad \text{(MWPM-P)}$$
$$\sum_{e \in N(i)} x_e = 1 \qquad \forall i \in V$$
$$x_e \geq 0 \qquad \forall e \in E$$

$$\max \quad \sum_{i \in V} y_i \qquad \text{(MWPM-D)}$$
$$y_i + y_j \leq c_e \quad \forall e = ij \in E$$

We now consider the problem from another angle, factoring in learning aspects. Suppose the graph $G = (V, E)$ is fixed but the edge cost vector $c \in \mathbb{Z}_+^E$ varies (is drawn from some distribution $\mathcal{D}$). If we are given an optimal dual $y^*$ as a prediction, then we can solve the problem by solving the max cardinality matching problem only once. However, the optimal dual can significantly change depending on edge cost $c$. Nevertheless, we will show how to learn "good" dual values and use them later to solve new MWPM instances faster. Specifically, we seek to design an end-to-end algorithm addressing all the aforementioned challenges:

1. **Feasiblity** (Section 3.1). The learned dual $\hat{y}$ may not be feasible for MWPM-D with some specific cost vector $c$. We show how to quickly convert it to a feasible dual $\hat{y}'(c)$ by appropriately decreasing the dual values (the more we decrease them, the further we move away from the optimum). Finding the feasible dual minimizing $\|\hat{y} - \hat{y}'(c)\|_1$ turns out to be a variant of the vertex cover problem, for which we give a simple 2-approximation running in $O(m + n)$ time. As a result, we have $\|\hat{y}'(c) - y^*(c)\|_1 \leq 3\|\hat{y} - y^*(c)\|_1$. See Theorem 4.
2. **Optimization** (Section 3.2). Now that we have a feasible solution $\hat{y}'(c)$, we want to find an optimal solution starting with $\hat{y}'(c)$ in time that depends on the quality of $\hat{y}'(c)$. Fortunately, the Hungarian algorithm can be seeded with any feasible dual, so we can "warm-start" it with $\hat{y}'(c)$. We show that its running time will be proportional to $\|\|\hat{y}'(c)\|_1 - \|y^*(c)\|_1\| \leq \|\hat{y}'(c) - y^*(c)\|_1$. Our analysis does not depend on the details of the Hungarian algorithm, and so applies to a broader class of primal-dual algorithms.
3. **Learnability** (Section 3.3). The target dual we seek to learn is $\arg\min_y \mathbb{E}_{c \sim \mathcal{D}} \|y - y^*(c)\|$; here $y^*(c)$ is the optimal dual for MWPM-D with cost vector $c$. We show we can efficiently learn $\hat{y}$ that is arbitrarily close to the target vector after $\tilde{O}(C^2 n^3)$ samples from $\mathcal{D}$. See Theorem 6.

Combining all of these gives the following, which is a more formal version of Theorem 1. Let $\mathcal{D}$ be an arbitrary distribution over edge costs where every vector in the support of $\mathcal{D}$ has maximum cost $C$. For any edge cost vector $c$, let $y^*(c)$ denote the optimal dual solution.

**Theorem 3.** *For any $p, \epsilon > 0$, there is an algorithm which:*

- *After $O\left(\left(\frac{nC}{\epsilon}\right)^2 (n \log n + \log(1/p))\right)$ samples from $\mathcal{D}$, returns dual values $\hat{y}$ such that $\mathbb{E}_{c \sim \mathcal{D}}[\|\hat{y} - y^*(c)\|_1] \leq \min_y \mathbb{E}_{c \sim \mathcal{D}}[\|y - y^*(c)\|_1] + \epsilon$ with probability at least $1 - p$.*
- *Using the learned dual $\hat{y}$, given edge costs $c$, computes a min-cost perfect matching in time $O\left(m\sqrt{n} \cdot \min\{\|\hat{y} - y^*(c)\|_1, \sqrt{n}\}\right)$.*

In the rest of this section we detail our proof of this Theorem.

## 3.1 Recovering a Feasible Dual Solution (Feasibility)

Let $\hat{y}$ be an infeasible set of (integral) dual prices – this should be thought of as the "good" dual obtained by our learning algorithm. Our goal in this section is to find a new *feasible* dual solution $\hat{y}'(c)$ that is close to $\hat{y}$, for a given MWPM-D instance with cost $c$. In particular we seek to find the closest feasible dual under the $\ell_1$ norm, i.e. one minimizing $\|\hat{y}'(c) - \hat{y}\|_1$.

---
**Algorithm 1** Fast Approx. for Distance to Feasibility

1: **procedure** FASTAPPROX($G = (V, E), r$)
2:      $\forall i \in V, \delta_i \leftarrow 0$
3:      **while** $E \neq \emptyset$ **do**
4:          Let $i$ be an arbitrary vertex of $G$
5:          **while** $i$ has a neighbor **do**
6:              $j \leftarrow \arg\max_{j' \in N(i)} r_{ij'}$
7:              $\delta_i \leftarrow r_{ij}$
8:              Delete $i$ and all its edges from $G$
9:              $i \leftarrow j$
10:     Return $\delta$

---

Looking at (MWPM-D), it is clear that we need to decrease the given dual values $\hat{y}$ in order to make it feasible. More formally, we are looking for a vector of non-negative perturbations $\delta$ such that $\hat{y}' := \hat{y} - \delta$ is feasible. We model finding the best set of perturbations, in terms of preserving $\hat{y}$'s dual objective value, as a linear program. Let $F := \{e = ij \in E \mid \hat{y}_i + \hat{y}_j > c_{ij}\}$ be the set of dual infeasible edges under $\hat{y}$. Define $r_e := \hat{y}_i + \hat{y}_j - c_e$ for each edge $e = ij \in F$. Asserting that $\hat{y} - \delta$ is feasible for (MWPM-D) while minimizing the amount lost in the dual objective leads to the following linear program:

$$\min \sum_{i \in V} \delta_i; \quad \delta_i + \delta_j \geq r_{ij} \ \ \forall ij \in F; \quad \delta_i \geq 0 \ \ \forall i \in V \tag{1}$$

Note that this is a variant of the vertex cover problem—the problem becomes exactly the vertex cover problem if $r_{ij} = 1$ for all edges $ij$. We could directly solve this linear program, but we are interested in making this step efficient. To find a fast approximation for (1), we take a simple greedy approach.

Algorithm 1 is a modification of the algorithm of Drake and Hougardy [15] which walks through the graph setting $\delta_i$ appropriately at each step to satisfy the covering constraints in (1). The analysis is based on interpreting the algorithm through the lens of primal-dual—the dual of (1) turns out to be a maximum weight matching problem with new edge weights $r_{ij}$.

A similar analysis as in Drake and Hougardy [15] implies this is a 2-approximation which runs in $O(m + n)$ time (all proofs are in the Supplementary material). This essentially immediately implies the following theorem (the 2-approximation turns into 3 due to a use of the triangle inequality).

**Theorem 4.** *There is a $O(m+n)$ time algorithm that takes an infeasible integer dual $\hat{y}$ and constructs a feasible integer dual $\hat{y}'(c)$ for MWPM-D with cost vector $c$ such that $\|\hat{y}'(c) - \hat{y}\|_1 \leq 2\|\hat{y} - y^*(c)\|_1$ where $y^*(c)$ is the optimal dual solution for MWPM-D with cost vector $c$. Thus by triangle inequality we have $\|\hat{y}'(c) - y^*(c)\|_1 \leq 3\|\hat{y} - y^*(c)\|_1$.*

## 3.2 Seeding Hungarian with a Feasible Dual (Optimization)

In this section we assume that we are given a feasible integral dual $\hat{y}'(c)$ for an input with cost vector $c$ and the goal is to find an optimal solution. We want to analyze the running time in terms of $\|\hat{y}'(c) - y^*(c)\|_1$, the distance to optimality. We use a simple primal-dual schema to achieve this, which is given formally in Algorithm 2.

To satisfy complementary slackness, we must only choose edges with $y_i + y_j = c_{ij}$. Let $E'$ be the set of such edges. We find a maximum cardinality matching in the graph $G' = (V, E')$. If the resulting matching $M$ is perfect then we are done by complementary slackness (Lemma 2) Otherwise, in steps 7-9 we modify the dual in a way that guarantees a strict increase in the dual objective. Since all parameters of the problem are integral, this strict increase then implies our desired bound on the number of iterations.

We will show this algorithm performs at most $O(\|y^*(c) - \hat{y}'(c)\|_1)$ iterations. We can further improve this by ensuring the algorithm runs no longer than the standard Hungarian algorithm in the case that we have large error in the prediction, i.e., $\|y^*(c) - \hat{y}'(c)\|_1$ is large. In particular, steps 6 and 11 do not precisely specify the choice of the set $S$ and the matching $M$. If we instantiate these steps appropriately (let $S = L \setminus C$ for step 6, where $C$ is a minimum vertex cover, and update $M$ along shortest-augmenting-paths for step 11) then we recover the Hungarian Algorithm. Together, we can prove the following theorem.

**Theorem 5.** *For an arbitrary cost vector $c$, the Hungarian method starting with a feasible integer dual solution $\hat{y}'(c)$ finds a minimum weight perfect matching in $\tilde{O}\left(m\sqrt{n} \cdot \min\{\|y^*(c) - \hat{y}'(c)\|_1, \sqrt{n}\}\right)$ time, where $y^*(c)$ is an optimal dual solution.*

## 3.3 Learning Optimal Advice (Learning)

Now we want to formally instantiate the "learning" part of our framework: if there is a good starting dual solution for a given input distribution, we want to find it without seeing too many samples. The formal model we will use is derived from data driven algorithm design and PAC learning.

We imagine solving many problem instances drawn from the same distribution. To formally model this, we let $\mathcal{D}$ be an unknown distribution over instances. For simplicity, we consider the graph $G = (V, E)$ to be fixed with vary-

---

**Algorithm 2** Simple Primal-Dual Scheme for MWPM

1: **procedure** MWPM-PD($G = (V, E), c, y$)
2:     $E' \leftarrow \{e \in E \mid y_i + y_j = c_{ij}\}$  ▷ Tight Edges
3:     $G' \leftarrow (V, E')$  ▷ $G$ containing only tight edges
4:     $M \leftarrow$ Maximum cardinality matching in $G'$
5:     **while** $M$ is not a perfect matching **do**
6:         Find $S \subseteq L$ such that $|S| > |\Gamma(S)|$ in $G'$
7:         $\epsilon \leftarrow \min_{i \in S, j \in R \setminus \Gamma(S)}\{c_{ij} - y_i - y_j\}$
8:         $\forall i \in S, y_i \leftarrow y_i + \epsilon$
9:         $\forall j \in \Gamma(S), y_j \leftarrow y_j - \epsilon$
10:         Update $E', G'$
11:         $M \leftarrow$ Maximum cardinality matching in $G'$
12:     Return $M$

---

ing costs. Thus $\mathcal{D}$ is a distribution over cost vectors $c \in \mathbb{R}^E$. We assume that the costs in this distribution are bounded. Let $C := \max_{c \sim \mathcal{D}} \max_{e \in E} c_e$ be finite and known to the algorithm. Our goal is to find the (not necessarily feasible) dual assignment that performs "best" in expectation over the distribution. Based on Theorems 4 and 5, we know that the "cost" of using dual values $y$ when the optimal dual is $y^*$ is bounded by $O(m\sqrt{n}\|y^* - y\|_1)$, and hence it is natural to define the "cost" of $y$ as $\|y^* - y\|_1$.

For every $c \in \mathbb{R}^E$ we will let $y^*(c)$ be a fixed optimal dual solution for $c$:

$$y^*(c) := \arg\max_y \left\{ \sum_i y_i \mid \forall ij \in E, y_i + y_j \leq c_{ij} \right\}.$$

Here we assume without loss of generality that $y^*(c)$ is integral as the underlying polytope is known to be integral. We will let the loss of a dual assignment $y$ be its $\ell_1$-distance from the optimal solution:

$$L(y, c) = \|y - y^*(c)\|_1.$$

Our goal is to learn dual values $\hat{y}$ which minimize $\mathbb{E}_{c \sim \mathcal{D}}[L(y, c)]$. Let $y^*$ denote the vector minimizing this objective, $y^* = \arg\min_y \mathbb{E}_{c \sim \mathcal{D}}[L(y, c)]$.

We will give PAC-style bounds, showing that we only need a small number of samples in order to have a good probability of learning an approximately-optimal solution $\hat{y}$. Our algorithm is conceptually quite simple: we minimize the empirical loss after an appropriate number of samples. In the supplementary we show that this can be done efficiently, giving the following theorem.

**Theorem 6.** *There is an algorithm that after $s = O\left(\left(\frac{nC}{\epsilon}\right)^2 (n \log n + \log(1/p))\right)$ samples returns dual values $\hat{y}$ such that $\mathbb{E}_{c \sim \mathcal{D}}[L(\hat{y}, c)] \leq \mathbb{E}_{c \sim \mathcal{D}}[L(y^*, c)] + \epsilon$ with probability at least $1 - p$. The algorithm runs in time polynomial in $n, m$ and $s$.*

This theorem, together with Theorems 4 and 5, immediately implies Theorem 3.

At the heart of the proof of Theorem 6 lies the proof of the following theorem regarding pseudo-dimensions of $\ell_1$-norm distance functions, which may be of independent interest.

**Theorem 7.** *Let $\mathcal{H}_n = \{f_y \mid y \in \mathbb{R}^n\}$ where $f_y : \mathbb{R}^n \to \mathbb{R}$ is defined by $f_y(x) = \|y - x\|_1$. The pseudo-dimension of $\mathcal{H}_n$ is at most $O(n \log n)$.*

We remark that minimizing this emprical loss can be efficiently implemented by taking the coordinate-wise median of each optimal dual, i.e. taking $y_j = \mathrm{median}(x_j^1, x_j^2, \ldots, x_j^s)$ for each $j \in V$.

| Dataset | Blog Feedback [10] | Covertype | KDD | Skin [9] | Shuttle |
|---|---|---|---|---|---|
| # of Points ($n$) | 52,397 | 581,012 | 98,942 | 100,000 | 43500 |
| # of Features ($d$) | 281 | 54 | 38 | 4 | 10 |

Table 1: Datasets used in experiments based on Euclidean data

## 4   Experiments

In this section we present experimental results on both synthetic and real data sets. Our goal is to validate the two main hypotheses in this work. First we show that warm-starting the Hungarian algorithm with learned duals provides an empirical speedup. Next, we show that the sample complexity of learning good duals is small, ensuring that our approach is viable in practice. We present some representative experimental results here; additional results are in the Supplementary Material.

**Experiment Setup:** All of our experiments were run on Google Cloud Platform [24] `e2-standard-2` virtual machines with 2 virtual CPU's and 8 GB of memory.

We consider two different setups for learning dual variables and evaluating our algorithms.

- Batch: In this setup, we receive $s$ samples $c_1, c_2, \ldots, c_s$ from the distribution of problem instances, learn the appropriate dual variables, and then test on new instances drawn from the distribution.
- Online: A natural use case for our approach is an *online* setting, where instance graphs $G_1, G_2, \ldots$ arrive one at a time. When deciding on the best warm start solution for $G_t$ we can use all of the data from $G_1, \ldots, G_{t-1}$. This is a standard scenario in industrial applications like ad matching, where a new ad allocation plan may need to be computed daily or hourly.

**Datasets:** To study the effect of the different algorithm parameters, we first run a study on synthetic data. Let $n$ be the number of nodes on one side of the bipartition and let $\ell, v$ be two parameters we set later. First, we divide the $n$ nodes on each side of the graph into $\ell$ groups of equal size. The weight of all edges going from the $i$'th group on the left side and the $j$'th group on the right side is initialized to some value $W_{i,j}$ drawn from a geometric distribution with mean 250. Then to generate a particular graph instance, we perturb each edge weight with independent random noise according to a binomial distribution, shifted and scaled so that it has mean 0 and variance $v$. We refer to this as the *type* model (each type consists of a group of nodes). We use $n = 500$, $\ell \in \{50, 100\}$ and vary $v$ from 0 to $2^{20}$.

We use the following model of generating instances from real data. Let $X$ be a set of $n$ points in $\mathbb{R}^d$, and fix a parameter $k$. We first divide $X$ randomly into two sets, $X_L$ and $X_R$ and compute a $k$-means clustering on each partition. To generate an instance $G = (L \cup R, E)$, we sample one point from each cluster on each side, generating $2k$ points in total. The points sampled from $X_L$ (resp. $X_R$) form the vertices in $L$ (resp. $R$). The weight of an $(i, j)$ edge is the Euclidean distance between these two points. Changing $k$ allows us to control the size of the instance.

We use several datasets from the UCI Machine Learning repository [16]. See Table 1 for a summary. For the KDD and Skin datasets we used a sub-sample of the original data (sizes given in Table 1).

**Implemented Algorithms and Metrics:** We implemented the Hungarian Algorithm (a particular instantiation of Algorithm 2, as discussed in Section 3.2) allowing for arbitrary seeding of a feasible integral dual. We experimented with having initial dual of 0 (giving the standard Hungarian Algorithm) as the baseline and having the initial duals come from our learning algorithm followed by Algorithm 1 to ensure feasibility (which we refer to as "Learned Duals"). We also added the following "tightening" heuristic, which is used in all standard implementations of the Hungarian algorithm: given any feasible dual solution $y$, set $y_i \leftarrow y_i + \min_{j \in N(i)} \{c_{ij} - y_i - y_j\}$ for all nodes $i$ on one side of the bipartition. This can be quickly carried out in $O(n + m)$ time, and guarantees that each node on that side has at least one edge in $E'$. We compare the runtime and number of primal-dual iterations, reporting mean values and error bars denoting 95% confidence intervals. The runtime results are in the supplementary material, and exhibit similar behavior (i.e., the extra running time caused by using Algorithm 1 in Learned Duals is negligible).

To learn initial duals we use a small number of independent samples of each instance type. We compute an optimal dual solution for each instance in the sample. To combine these together into

a single dual solution, we compute the median value for each node's set of dual values. This is an efficient implementation of the empirical risk minimization algorithm from Section 3.3.

**Results:** First, we examine the performance of Learned Duals in the batch setting described above. For these experiments, we used 20 training instances to learn the initial duals and then tested those on 10 new instances. For the type model, we used $\ell = 50$ and considered varying the variance parameter $v$. The left plot in Figure 1 shows the results as we increase $v$ from 0 to 300. We see a moderate improvement in this case, even when the noise variance is larger than the mean value of an edge weight. Going further, in the middle plot of Figure 1 we consider increasing the noise variance in powers of two geometrically. Note that even when the noise significantly dominates the original signal from the mean weights (and hence the training instances should not help on the test instances), our method is comparable to the Hungarian method.

Continuing with the Batch setting, the right plot in Figure 1 summarizes our results for the clustering derived instances on all datasets with $k = 500$ (similar results hold for other values of $k$; see the Supplementary Materials). We see an improvement across all datasets, and a greater than 2x improvement on all but the Covertype dataset.

Figure 2 has three plots on the online setting. We aim to show that not too many samples are needed to learn effective duals. From left to right, the plots in Figure 2 show the performance averaged over 20 repetitions of the experiment with 20 time points on the type model with $\ell = 100, v = 200$, and the clustering derived instances on the KDD and Covertype datasets with $k = 500$, respectively. We see that only a few iterations are needed to see a significant separation between the run time of our method with learned duals and the standard Hungarian method, with further steady improvement as we see more instances. Additional plots for the other datasets are in the Supplementary Materials.

We see similar trends in both the real and synthetic data sets. We conclude the following.

- The theory is predictive of practice. Empirically, learning dual variables can lead to significant speed-up. This speed-up is achieved in both the batch and online settings.
- As the distribution is more concentrated, the learning algorithm performs better (as one would suspect).
- When the distribution is not concentrated and there is little to learn, then the algorithm has performance similar to the widely used Hungarian algorithm.

All together, these results demonstrate the strong potential for improvements in algorithm run time using machine-learned predictions for the weighted matching problem.

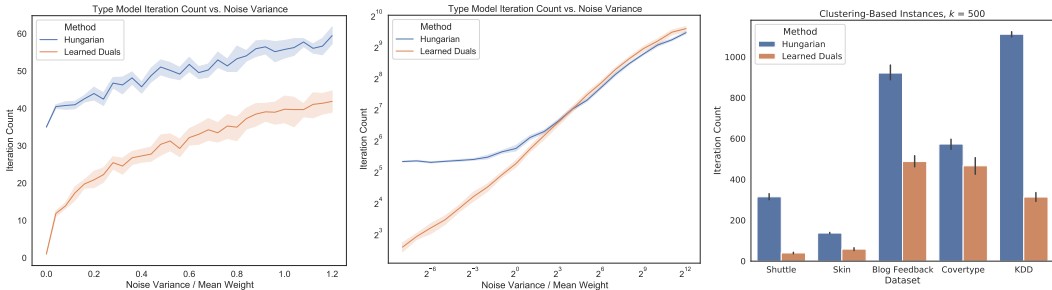

Figure 1: Iteration count results for the Batch setting. The left figure gives the iteration count for the type model (synthetic data) versus linearly increasing $v$, while the middle geometrically increases $v$. The right figure summarizes the results for clustering based instances (real data) in the batch setting.

## 5 Conclusion and Future Work

In this work we showed how to use learned predictions to warm-start primal-dual algorithms for weighted matching problems to improve their running times. We identified three key challenges of feasibility, learnability and optimization, for any such scheme, and showed that by working in the dual space we could give rigorous performance guarantees for each. Finally, we showed that our proposed methods are not only simpler, but also more efficient in practice.

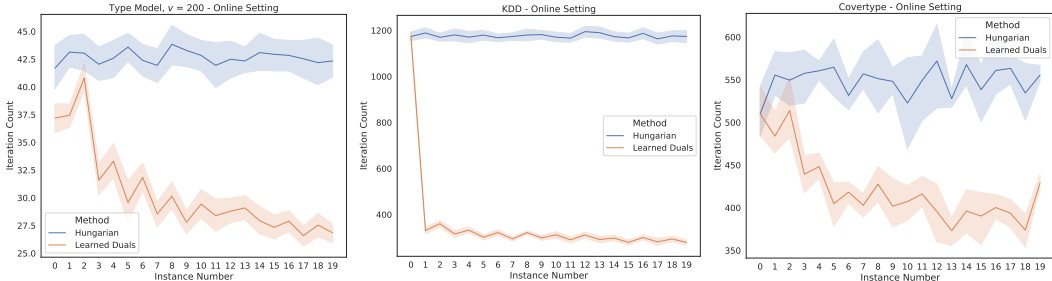

Figure 2: Iteration count results for the Online setting. The left figure is for the type model (synthetic data), while the middle and right are for the clustering based instances (real data) with $k = 500$ on KDD and Covertype, respectively.

An immediate avenue for future work is to extend these results to other combinatorial optimization problems. The key ingredient is identifying an appropriate intermediate representation: it must be simple enough to be learnable with small sample complexity, yet sophisticated enough to capture the underlying structure of the problem at hand.

## Acknowledgments and Disclosure of Funding

Michael Dinitz was supported in part by NSF grant CCF-1909111. Sungjin Im was supported in part by NSF grants CCF-1617653 and CCF-1844939. Thomas Lavastida and Benjamin Moseley were supported in part by NSF grants CCF-1824303, CCF-1845146, CCF-1733873 and CMMI-1938909. Benjamin Moseley was additionally supported in part by a Google Research Award, an Infor Research Award, and a Carnegie Bosch Junior Faculty Chair.

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
