The minimum cost $b$-matching problem and its generalization, the minimum cost flow problem, have also been extensively studied. See [44] for a summary of classical results. More recently there has been improvements by applying interior point methods. The algorithm of [15] has running time $\tilde{O}(m^{3/2} \log^2(C))$ and it is improved by the algorithm of [37] which runs in time $\tilde{O}(m\sqrt{n} \log^{O(1)}(C))$. We note that the recent breakthrough of van den Brand et al. [53] solves the minimum cost problem in time $\tilde{O}((m + n^{1.5})n^{o(1)} \log^2(BC))$, where $B$ is the maximum capacity and $C$ is the maximum edge weight.

Large scale bipartite matchings have been studied extensively in the online setting, as they represent the basic problem in ad allocations [39]. While the ad allocation is inherently online, most of the methods precompute a dual based solution based on a sample of the input [16, 55], and then argue that this solution is approximately optimal on the full instance. In contrast, we strive to compute the *exactly optimal* solution, but use previous instances to improve the running time of the approach.

**Algorithms with Predictions:** Kraska et al. [35] showed how to use machine learned predictions to improve heavily optimized indexing algorithms. The original paper was purely empirical and came with no rigorous guarantees; recently there has been a flurry of work putting such approaches on a strong theoretical foundation, evaluating the benefit of augmenting classical algorithms with machine learned predictions, see [41] for a survey. Online and streaming algorithms in particular have seen significant successes, as predictions reveal information about the future and can help guide the algorithms' choices. This has led to the design of new methods for caching [38, 49, 32], scheduling [48, 36], frequency counting [14, 31, 1], and membership testing [40, 52] that can break through worst-case lower bounds when the predictions are of sufficiently high quality.

Most of the above work abstracts the predictions as access to an error-prone oracle and asks how to best use predictions: getting performance gains when the predictions are good, but limiting the losses when they are not. A related emergent area is that of data driven algorithm design [28, 8, 6, 5, 7, 13]. Here, the objective is to "learn" a good algorithm for a particular family of inputs. The goal is not typically to tie the performance of the algorithm to the quality of the prediction, but rather to show that the prediction makes sense; that is only a small number of problem samples are needed in order to ensure the learned algorithm generalizes to new data points.

**Comparison to Dynamic Algorithms:** A natural counterpoint to our approach is the area of *dynamic algorithms*. In dynamic algorithms, we attempt to design algorithms that allow us to very quickly recompute the optimal solution when there is a single (or a very small number) of changes in the input. In other words, the input is changing over time, and we need to always maintain an optimal solution as it changes. There has been an active line of work on dynamic matching algorithms, see [11, 51].

This is in some sense very similar to what we are trying to do, since we are also trying to quickly compute an optimal solution when we have a history and previous optimal solutions. But these two approaches, of dynamic algorithms and of machine-learned predictions for warm-start, are quite different and are actually highly complementary. Dynamic algorithms work extremely well in the setting where the input changes slowly but where the output can change quickly, since they are optimized to handle *single* changes in the input (e.g., a single edge being added or removed from the graph). Our approach, on the other hand, works extremely well when the input can change dramatically but the optimal solution is relatively stable, since then our learned dual values will be quite close to optimal.

## 1.3 Roadmap

We begin with preliminaries and background in Section 2. We then present our main theoretical results on min-cost perfect bipartite matching in Section 3. The experiments are presented in Section 4. Finally, the extension to $b$-matching is presented in Section 5.

## 2 Preliminaries

**Notation:** Let $G = (V, E)$ be an undirected graph. When $G$ is bipartite we will use $L$ and $R$ to refer to the two sides of the bipartition. We will let $N(i) := \{e \in E \mid i \in e\}$ be the set of edges adjacent to vertex $i$. Similarly if $G$ is directed, then we use $N^+(i)$ and $N^-(i)$ to be the set of edges leaving $i$ and the set of edges entering $i$, respectively. For a set $S \subseteq V$, let $\Gamma(S)$ be the vertex

neighborhood of $S$. For a vector $y \in \mathbb{R}^n$, we let $\|y\|_1 = \sum_i |y_i|$ be its $\ell_1$-norm. Let $\langle x, y \rangle$ be the standard inner product on $\mathbb{R}^n$.

**Linear Programming and Complementary Slackness:** Here we recall optimality conditions for linear programming that are used to ensure the correctness of some algorithms we present. Consider the primal-dual pair of linear programs below.

$$
\begin{aligned}
\min \quad & c^\top x \\
& Ax = b \\
& x \geq 0
\end{aligned}
\tag{$P$}
$$

$$
\begin{aligned}
\max \quad & b^\top y \\
& A^\top y \leq c
\end{aligned}
\tag{$D$}
$$

A pair of solutions $x, y$ for $(P)$ and $(D)$, respectively, satisfy complementary slackness if $x^\top(c - A^\top y) = 0$. The following lemma is well-known.

**Lemma 2.** *Let $x$ be a feasible solution for $(P)$ and $y$ be a feasible solution for $(D)$. If the pair $x, y$ satisfies complementary slackness, then $x$ and $y$ are optimal solutions for their respective problems.*

**Maximum Cardinality Matching:** Let $G = (V, E)$ be a bipartite graph on $n$ vertices and $m$ edges. A matching $M \subseteq E$ is a collection of non-intersecting edges. The Hopcroft-Karp algorithm for finding a matching maximizing $|M|$ runs in time $O(\sqrt{n} \cdot m)$ [29], which is still state-of-the-art for general bipartite graphs. For moderately dense graphs, a recent result by van den Brand et al. [53] gives a better running time of $\tilde{O}(m + n^{1.5})$ (where $\tilde{O}$ hides polylogarithmic factors).

**Minimum Weight Perfect Matching (MWPM):** Again, let $G = (V, E)$ be a bipartite graph on $n$ vertices and $m$ with costs $c \in \mathbb{Z}_+^E$ on the edges, and let $C$ be the maximum cost. A matching $M$ is *perfect* if every vertex is matched by $M$. The objective of this problem is to find a perfect matching $M$ minimizing the cost $c(M) := \sum_{e \in M} c_e$.

When looking for optimal solutions we can assume that $G$ is a complete graph by adding all possible edges not in $E$ with weight $Cn^2$. It is easy to see that any $o(n)$ approximate solution would not use any of these edges.

**Maximum Flow:** Now let $G = (V, E)$ be a directed graph on $n$ vertices and $m$ edges with a capacity vector $u \in \mathbb{R}_+^E$. Let $s$ and $t$ be distinct vertices of $G$. An $st$-flow is a vector $f \in \mathbb{R}_+^E$ satisfying $\sum_{e \in N^+(i)} f_e - \sum_{e \in N^-(i)} f_e = 0$ for all vertices $i \neq s, t$. An $st$-flow $f$ is maximum if it maximizes $\sum_{e \in N^+(s)} f_e = \sum_{e \in N^-(t)} f_e$. The algorithm due to Orlin [45] and King, Rao, and Tarjan [34] runs in time $O(nm)$.

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

$$\delta_i + \delta_j \geq r_{ij} \quad \forall ij \in F$$
$$\delta_i \geq 0 \qquad \forall i \in V$$

(1)

Note that this is a variant of the vertex cover problem—the problem becomes exactly the vertex cover problem if $r_{ij} = 1$ for all edges $ij$. We could directly solve this linear program, but we are interested in making this step efficient. To find a fast approximation for (1), we take a simple greedy approach.

---

**Algorithm 1** Fast Approx. for Distance to Feasibility

---

1: **procedure** FASTAPPROX($G = (V, E), r$)
2:      $\forall i \in V, \delta_i \leftarrow 0$
3:      **while** $E \neq \emptyset$ **do**
4:          Let $i$ be an arbitrary vertex of $G$
5:          **while** $i$ has a neighbor **do**
6:              $j \leftarrow \arg\max_{j' \in N(i)} r_{ij'}$
7:              $\delta_i \leftarrow r_{ij}$
8:              $\gamma_{ij} = 1/2$              $\triangleright$ $\gamma_{ij}$ is only used for analysis
9:              Delete $i$ and all its edges from $G$
10:             $i \leftarrow j$
11:      Return $\delta$

---

Algorithm 1 is a modification of the algorithm of Drake and Hougardy [18] which walks through the graph setting $\delta_i$ appropriately at each step to satisfy the covering constraints in (1). The analysis is based on interpreting the algorithm through the lens of primal-dual—the dual of (1) turns out to be a maximum weight matching problem with new edge weights $r_{ij}$. The dual is the following:

$$\max \quad \sum_e r_e \gamma_e$$
$$\sum_{e \in N(i) \cap F} \gamma_e \leq 1 \quad \forall i \in V$$
$$\gamma_e \geq 0 \qquad \forall e \in F$$

(2)

First we show the algorithm is fast.

**Lemma 4.** *Algorithm 1 runs in time $O(n + m)$.*

*Proof.* This follows from the trivial observation that each vertex/edge is considered $O(1)$ times. $\square$

Next, by construction the algorithm constructs a feasible dual solution.

**Lemma 5.** *The perturbations $\delta$ returned by Algorithm 1 is feasible for* (1).

*Proof.* We want to show that $\delta_i + \delta_j \geq r_e$ for all edges $e = ij \in E$. We claim that this condition holds whenever the edge is deleted from $G$. Suppose that the algorithm is currently at $i$ and let $ij'$ be the edge selected by the algorithm in this step. By definition of the algorithm we have $\delta_i = r_{ij'} \geq r_{ij}$ so $\delta_i + \delta_j \geq r_{ij}$. $\square$

Finally, we address the objective.

**Lemma 6.** *The perturbations $\delta$ returned by Algorithm 1 are a 2-approximation for* (1).

*Proof.* In each iteration, the increase of the primal objective ($\delta_i = r_{ij}$ in Line 7) is exactly twice the increase of the dual objective ($r_{ij}\gamma_{ij} = r_{ij}/2$ in Line 8). Thus, due to weak duality, it suffices to show that the dual is feasible. This follows from the observation that $\{ij \mid \gamma_{ij} = 1/2\}$ forms a collection of vertex disjoint paths and cycles. Thus, for every $i \in V$, there are at most two edges $e$ adjacent to $i$ such that $\gamma_e > 0$, and for those edges $e$, $\gamma_e = 1/2$. Therefore, the dual is feasible. $\square$

This shows we can project the predicted dual prices $\hat{y}$ onto the set of feasible dual prices at approximately the minimum cost. The prior lemmas give the following theorem by noticing that $y^*(c)$ is a possible feasible solution. Note that integrality is immediate from the algorithm.

**Theorem 7.** *There is a $O(m+n)$ time algorithm that takes an infeasible integer dual $\hat{y}$ and constructs a feasible integer dual $\hat{y}'(c)$ for MWPM-D with cost vector $c$ such that $\|\hat{y}'(c) - \hat{y}\|_1 \leq 2\|\hat{y} - y^*(c)\|_1$ where $y^*(c)$ is the optimal dual solution for MWPM-D with cost vector $c$. Thus by triangle inequality we have $\|\hat{y}'(c) - y^*(c)\|_1 \leq 3\|\hat{y} - y^*(c)\|_1$.*

### 3.2 Seeding Hungarian with a Feasible Dual (Optimization)

In this section we assume that we are given a feasible integral dual $\hat{y}'(c)$ for an input with cost vector $c$ and the goal is to find an optimal solution. We want to analyze the running time in terms of $\|\hat{y}'(c) - y^*(c)\|_1$, the distance to optimality. We use a simple primal-dual schema to achieve this, which is given formally in Algorithm 2.

---

**Algorithm 2** Simple Primal-Dual Scheme for MWPM

1: **procedure** MWPM-PD($G = (V, E), c, y$)
2:      $E' \leftarrow \{e \in E \mid y_i + y_j = c_{ij}\}$          ▷ Set of tight edges in the dual
3:      $G' \leftarrow (V, E')$          ▷ $G$ containing only tight edges
4:      $M \leftarrow$ Maximum cardinality matching in $G'$
5:      **while** $M$ is not a perfect matching **do**
6:          Find $S \subseteq L$ such that $|S| > |\Gamma(S)|$ in $G'$          ▷ Exists by Hall's Theorem
                                                     ▷ Can be found in $O(m+n)$ time
7:          $\epsilon \leftarrow \min_{i \in S, j \in R \setminus \Gamma(S)} \{c_{ij} - y_i - y_j\}$
8:          $\forall i \in S, y_i \leftarrow y_i + \epsilon$
9:          $\forall j \in \Gamma(S), y_j \leftarrow y_j - \epsilon$
10:         Update $E', G'$
11:         $M \leftarrow$ Maximum cardinality matching in $G'$
12:      Return $M$

---

To satisfy complementary slackness, we must only choose edges with $y_i + y_j = c_{ij}$. Let $E'$ be the set of such edges. We find a maximum cardinality matching in the graph $G' = (V, E')$. If the resulting matching $M$ is perfect then we are done by complementary slackness (Lemma 2) Otherwise, in steps 7-9 we modify the dual in a way that guarantees a strict increase in the dual objective. Since all parameters of the problem are integral, this strict increase then implies our desired bound on the number of iterations.

We now analyze Algorithm 2. Recall that $L$ and $R$ give the bipartition of $V$. First we show that the algorithm is correct. The main claim we need to establish is that if $y$ is initially dual feasible, then it remains dual feasible throughout the algorithm. First we check that the update defined in lines 6-10 is well defined, i.e. in line 6 such a set $S$ always exists and $\epsilon$ defined in line 7 is always strictly positive.

**Proposition 8.** *If $M$ is not a perfect matching in $G'$, then there exists a set $S \subseteq L$ such that $|S| > |\Gamma(S)|$ in $G'$. Further, such $S$ can be found in $O(m+n)$ time.*

*Proof.* The first claim follows directly from Hall's Theorem applied to $G'$. It is well-known that the maximum matching size is equal to the minimum vertex cover size when the underlying graph is bipartite. Further, a minimum vertex cover $C$ can be derived from a maximum matching $M$ in time $O(m+n)$. We set $S = L \setminus C$. Then, we have $\Gamma(S) \subseteq C \cup R$ due to $C$ being a vertex cover, and $|C \cap L| + |C \cap R| = |C| < n$ as the minimum cover size is less than $n$; recall $M$ is not perfect. Thus, we have $|S| = n - |C \cap L| > |C \cap R| \geq |\Gamma(S)|$, as desired. $\square$

**Proposition 9.** *Let $y$ be dual feasible and suppose that $S \subseteq L$ with $|S| > |\Gamma(S)|$ in $G'$. Let $\epsilon = \min_{i \in S, j \in R \setminus \Gamma(S)} c_{ij} - y_i - y_j$. Then as long as $c$ and $y$ are integer we have $\epsilon \geq 1$.*

*Proof.* Every edge $ij$ considered in the definition of $\epsilon$ is not in $E'$ and thus must have $c_{ij} > y_i + y_j$. Thus for all such edges we have $c_{ij} - y_i - y_j \geq 1$ since $c$ and $y$ are integer, and so $\epsilon \geq 1$.

If no such edge exists, then we have a set $S \subseteq L$ such that $|S|$ is strictly larger than its neighborhood in $G$ (rather than $G'$) which shows that the problem is infeasible. This contradicts our assumption that the original problem is feasible. $\square$

We now show the main claims we described above.

**Lemma 10.** *If Algorithm 2 is given an initial dual feasible $y$, then $y$ remains dual feasible throughout its execution.*

*Proof.* Inductively, it suffices to show that if $y$ is dual feasible then it remains so after the update steps defined in lines 6-10. To make the notation clear, let $y'$ be the result of applying the update rule to $y$. Consider an edge $ij \in E$. We want to show that $y'_i + y'_j \le c_{ij}$ after the update step. There are 4 cases to check: (1) $i \in L \setminus S, j \in R \setminus \Gamma(S)$, (2) $i \in L \setminus S, j \in \Gamma(S)$, (3) $i \in S, j \in R \setminus \Gamma(S)$, and (4) $i \in S, j \in \Gamma(S)$.

In the first case, neither $y_i$ nor $y_j$ are modified, so we get $y'_i + y'_j = y_i + y_j \le c_{ij}$ since $y$ was initially dual feasible. In the second case we have $y'_i + y'_j = y_i + y_j - \epsilon \le c_{ij}$ since $\epsilon > 0$. In the third case we have $y'_i = y_i + \epsilon$ and so $y'_i + y'_j = y_i + \epsilon + y_j \le c_{ij}$ since there was slack on these edges and $\epsilon$ was chosen to be the smallest such slack. Finally, in the last case we have $y'_i + y'_j = y_i + \epsilon + y_j - \epsilon \le c_{ij}$. Thus we conclude that $y$ remains feasible throughout the execution of Algorithm 2. $\square$

**Lemma 11.** *Each iteration strictly increases the value of the dual solution.*

*Proof.* Note that in each iteration $y_i$ increases by $\epsilon$ for all $i \in S$ and $y_j$ decreases by $\epsilon$ for all $j \in \Gamma(S)$; and all other dual variables remain unchanged. Thus, the dual objective increases by $\epsilon(|S| - |\Gamma(S)|) \ge \epsilon$. $\square$

The above lemma allows us to analyze the running time of our algorithm in terms of the distance to optimality.

**Lemma 12.** *Consider an arbitrary cost vector $c$. Suppose that $\hat{y}'(c)$ is an integer dual feasible solution and $y^*(c)$ is an integer optimal dual solution. If Algorithm 2 is initialized with $\hat{y}'(c)$, then the number of iterations is bounded by $\|\hat{y}(c) - y^*(c)\|_1$.*

*Proof.* By Lemma 11, we have that the value of the dual solution increases by at least 1 in each iteration. Thus the number of iterations is at most $\sum_i y_i^*(c) - \sum_i \hat{y}_i(c) \le \sum_i |y_i^*(c) - \hat{y}_i(c)| = \|y^*(c) - \hat{y}(c)\|_1$. $\square$

Finally, we get the following theorem as a corollary of the lemmas above and the $O(m\sqrt{n})$ runtime of the Hopcroft-Karp algorithm for maximum cardinality matching [29]. More precisely, the above lemmas show that the algorithm performs at most $O(\|y^*(c) - \hat{y}'(c)\|_1)$ iterations, each running in $O(m\sqrt{n})$ time. We can further improve this by ensuring the algorithm runs no longer than the standard Hungarian algorithm in the case that we have large error in the prediction, i.e., $\|y^*(c) - \hat{y}'(c)\|_1$ is large. In particular, steps 6 and 11 do not precisely specify the choice of the set $S$ and the matching $M$. If we instantiate these steps appropriately (let $S = L \setminus C$ for step 6, where $C$ is a minimum vertex cover, and update $M$ along shortest-augmenting-paths for step 11) then we recover the Hungarian Algorithm and its $\tilde{O}(mn)$ running time.

**Theorem 13.** *Consider an arbitrary cost vector $c$. There exists an algorithm which takes as input a feasible integer dual solution $\hat{y}'(c)$ and finds a minimum weight perfect matching in $\tilde{O}\left(\min\{m\sqrt{n}\|y^*(c) - \hat{y}'(c)\|_1, mn\}\right)$ time, where $y^*(c)$ is an optimal dual solution.*

### 3.3 Learning Optimal Advice (Learning)

Now we want to formally instantiate the "learning" part of our framework: if there is a good starting dual solution for a given input distribution, we want to find it without seeing too many samples. The formal model we will use is derived from data driven algorithm design and PAC learning.

We imagine solving many problem instances drawn from the same distribution. To formally model this, we let $\mathcal{D}$ be an unknown distribution over instances. For simplicity, we consider the graph $G = (V, E)$ to be fixed with varying costs. Thus $\mathcal{D}$ is a distribution over cost vectors $c \in \mathbb{R}^E$.

We assume that the costs in this distribution are bounded. Let $C := \max_{c \sim \mathcal{D}} \max_{e \in E} c_e$ be finite and known to the algorithm. Our goal is to find the (not necessarily feasible) dual assignment that performs "best" in expectation over the distribution. Based on Theorems 7 and 13 , we know that the "cost" of using dual values $y$ when the optimal dual is $y^*$ is bounded by $O(m\sqrt{n}\|y^* - y\|_1)$, and hence it is natural to define the "cost" of $y$ as $\|y^* - y\|_1$.

For every $c \in \mathbb{R}^E$ we will let $y^*(c)$ be a fixed optimal dual solution for $c$:

$$y^*(c) := \arg\max_y \left\{ \sum_i y_i \mid \forall ij \in E, y_i + y_j \leq c_{ij} \right\}.$$

Here we assume without loss of generality that $y^*(c)$ is integral as the underlying polytope is known to be integral. We will let the loss of a dual assignment $y$ be its $\ell_1$-distance from the optimal solution:

$$L(y, c) = \|y - y^*(c)\|_1.$$

Our goal is to learn dual values $\hat{y}$ which minimize $\mathbb{E}_{c \sim \mathcal{D}}[L(y, c)]$. Let $y^*$ denote the vector minimizing this objective, $y^* = \arg\min_y \mathbb{E}_{c \sim \mathcal{D}}[L(y, c)]$.

We will give PAC-style bounds, showing that we only need a small number of samples in order to have a good probability of learning an approximately-optimal solution $\hat{y}$. Our algorithm is conceptually quite simple: we minimize the empirical loss after an appropriate number of samples. We have the following theorem.

**Theorem 14.** *There is an algorithm that after $s = O\left( \left(\frac{nC}{\epsilon}\right)^2 (n \log n + \log(1/p)) \right)$ samples returns dual values $\hat{y}$ such that $\mathbb{E}_{c \sim \mathcal{D}}[L(\hat{y}, c)] \leq \mathbb{E}_{c \sim \mathcal{D}}[L(y^*, c)] + \epsilon$ with probability at least $1 - p$. The algorithm runs in time polynomial in $n, m$ and $s$.*

This theorem, together with Theorems 7 and 13, immediately implies Theorem 3.

### 3.3.1 Proof of Theorem 14

We now discuss the main tools we require from statistical learning theory in order to prove Theorem 14. For every dual assignment $y \in \mathbb{R}^V$, we define a function $g_y : \mathbb{R}^E \to \mathbb{R}$ by $g_y(c) = L(y, c) = \|y^*(c) - y\|_1$. Let $\mathcal{H} = \{g_y \mid y \in \mathbb{R}^V\}$ be the collection of all such functions. It turns out that in order to prove Theorem 14, we just need to bound the *pseudo-dimension* of this collection. Note that the notion of shattering and pseudo-dimension in the following is a generalization to real-valued functions of the classical notion of VC-dimension for boolean-valued functions (classifiers).

**Definition 15.** *[47, 3, 42] Let $\mathcal{F}$ be a class of functions $f : X \to \mathbb{R}$. Let $S = \{x_1, x_2, \ldots, x_s\} \subset X$. We say that that $S$ is shattered by $\mathcal{F}$ if there exist real numbers $r_1, \ldots, r_s$ so that for all $S' \subseteq S$, there is a function $f \in \mathcal{F}$ such that $f(x_i) \leq r_i \iff x_i \in S'$ for all $i \in [s]$. The pseudo-dimension of $\mathcal{F}$ is the largest $s$ such that there exists an $S \subseteq X$ with $|S| = s$ that is shattered by $\mathcal{F}$.*

The connection between pseudo-dimension and learning is given by the following uniform convergence result.

**Theorem 16.** *[47, 3, 42] Let $\mathcal{D}$ be a distribution over a domain $X$ and $\mathcal{F}$ be a class of functions $f : X \to [0, H]$ with pseudo-dimension $d_\mathcal{F}$. Consider $s$ independent samples $x_1, x_2, \ldots, x_s$ from $\mathcal{D}$. There is a universal constant $c_0$, such that for any $\epsilon > 0$ and $p \in (0, 1)$, if $s \geq c_0 \left(\frac{H}{\epsilon}\right)^2 (d_\mathcal{F} + \ln(1/p))$ then we have*

$$\left| \frac{1}{s} \sum_{i=1}^s f(x_i) - \mathbb{E}_{x \sim \mathcal{D}}[f(x)] \right| \leq \epsilon$$

*for all $f \in \mathcal{F}$ with probability at least $1 - p$.*

Intuitively, this theorem says that the sample average $\frac{1}{s} \sum_{i=1}^s f(x_i)$ is close to its expected value for every function $f \in \mathcal{F}$ simultaneously with high probability so long as the sample size $s$ is large enough. This theorem can be utilized to give a learning algorithm for our problem by considering an algorithm which minimizes the empirical loss. In general, the "best" function is the one which minimizes the expected value over $\mathcal{D}$, i.e. $f^* = \arg\min_{f \in \mathcal{F}} \mathbb{E}_{x \sim \mathcal{D}}[f(x)]$. We have the following simple corollary for learning and approximately best function $\hat{h}$.

**Corollary 17.** *Consider a set of $s$ independent samples $x_1, x_2, \ldots, x_s$ from $\mathcal{D}$ and let $\hat{f}$ be a function in $\mathcal{F}$ which minimizes $\frac{1}{s} \sum_{i=1}^{s} \hat{f}(x_i)$. If $s$ is chosen as in Theorem 16, then with probability $1 - p$ we have $\mathbb{E}_{x \sim \mathcal{D}}[\hat{f}(x)] \leq \mathbb{E}_{x \sim \mathcal{D}}[f^*(x)] + 2\epsilon$*

Thus based on the above Theorem and Corollary, to prove Theorem 14 we must accomplish the following tasks. First and foremost, we must bound the pseudo-dimension of our class of functions $\mathcal{H}$. Next, we need to check that the functions are bounded on the domain we consider, and finally we need to give an algorithm minimizing the empirical risk. The latter two tasks are simple. By assumption the edge costs are bounded by $C$. If we restrict $\mathcal{H}$ to be within a suitable bounding box, then one can verify that we can take $H = O(nC)$ to satisfy the conditions for Theorem 16. Additionally, the task of finding a function to minimize the loss on the sample can be done via linear programming. We formally verify these details in Sections 3.3.2 and 3.3.3. This leaves bounding the pseudo-dimension of the class $\mathcal{H}$, which we focus on now.

To bound the pseudo-dimension of $\mathcal{H}$, we will actually consider a different class of functions $\mathcal{H}_n$: for every $y \in \mathbb{R}^n$ we define a function $f_y : \mathbb{R}^n \to \mathbb{R}$ by $f_y(x) = \|y - x\|_1$, and we let $\mathcal{H}_n = \{f_y \mid y \in \mathbb{R}^n\}$. It is not hard to argue that it is sufficient to bound the pseudo-dimension of this class.

**Lemma 18.** *If the pseudo-dimension of $\mathcal{H}_n$ is at most $k$, then the pseudo-dimension of $\mathcal{H}$ is at most $k$.*

*Proof.* We prove the contrapositive: we start with a set of size $s$ which is shattered by $\mathcal{H}$, and use it to find a set of size $s$ which is shattered by $\mathcal{H}_n$. Let $S = \{c_1, c_2, \ldots, c_s\}$ with each $c_i \in \mathbb{R}^E$ be a set which is shattered by $\mathcal{H}$. Then there are real numbers $r_1, r_2, \ldots, r_s$ so that for all $S' \subseteq [s]$, there is a function $g \in \mathcal{H}$ where $g(c_i) \leq r_i \iff i \in S'$. By definition of $\mathcal{H}$, this $g$ is $g_{y_{S'}}$ for some $y_{S'} \in \mathbb{R}^n$, and so $\|y_{S'} - y^*(c_i)\|_1 \leq r_i \iff i \in S'$.

Let $\hat{S} = \{y^*(c_1), y^*(c_2), \ldots, y^*(c_s)\}$. We claim that $\hat{S}$ is shattered by $\mathcal{H}_n$. To see this, consider the same real numbers $r_1, \ldots, r_s$ and some $S' \subseteq [s]$. Then $f_{y_{S'}}(y^*(c_i)) = \|y_{S'} - y^*(c_i)\|_1 = g_{y_{S'}}(c_i)$ and hence $f_{y_{S'}}(y^*(c_i)) \leq r_i \iff g_{y_{S'}}(c_i) \leq r_i \iff i \in S'$. Thus $\hat{S}$ is shattered by $\mathcal{H}_n$. $\square$

So now our goal is to prove the following bound, which (with Lemma 18 and Theorem 16) implies Theorem 14.

**Theorem 19.** *The pseudo-dimension of $\mathcal{H}_n$ is at most $O(n \log n)$.*

Let $k$ be the pseudo-dimension of $\mathcal{H}_n$. Then by the definition of pseudo-dimension there is a set $P = \{x^1, x^2, \ldots, x^k\}$ which is shattered by $\mathcal{H}_n$, so there are values $r_1, r_2, \ldots, r_k \in \mathbb{R}_{\geq 0}$ so that for all $S \subseteq P$ there is an $f \in \mathcal{H}_n$ such that $f(x^i) \leq r_i \iff x^i \in S$. By our definition of $\mathcal{H}_n$, this means that there is a $y_S \in \mathbb{R}^n$ so that $\|y_S - x^i\|_1 \leq r_i \iff x^i \in S$.

For each $S \subseteq P$, define the *region* of $S$ (denoted by $r(S)$) to be

$$r(S) = \{y \in \mathbb{R}^n : \|y - x^i\|_1 \leq r_i \iff x^i \in S\},$$

i.e., the set of points that are at $\ell_1$-distance at most $r_i$ from $x^i$ for precisely the $x^i$'s that are in $S$. Clearly each $r(S)$ is nonempty for every $S \subseteq P$ due to the existence of $y_S$. Let $m = 2^k$ be the number of nonempty regions.

To upper bound the pseudo-dimension $k$ we will prove that there cannot be too many nonempty regions (i.e., $m$ is small). This is somewhat complex since the $\ell_1$-balls have complex structure (in particular, have $2^d$ facets), so we will do this by partitioning $\mathbb{R}^n$ into *cells* in which the $\ell_1$ balls are simpler. For each $x^i \in P$ and $j \in [n]$, let $Q^i_j$ be the hyperplane in $\mathbb{R}^n$ that passes through $y_i$ and is perpendicular to the axis $e_j$ (i.e., $Q^i_j = \{y \in \mathbb{R}^n : \langle y - x^i, e_j \rangle = 0\}$). Clearly there are $kn$ of these hyperplanes. Define a *cell* to be a maximal set of points in $\mathbb{R}^n$ which are the same side of every hyperplane. Note that there are $(k+1)^n$ of these cells, they partition $\mathbb{R}^n$, and every cell which is bounded is a hypercube.

**Lemma 20.** *Let $C$ be a cell and $x^i \in P$. There is a halfspace $H$ such that $B_1(x^i, r_i) \cap C = H \cap C$.*

*Proof.* If $B_1(x^i, r_i) \cap C = \emptyset$ then we are done. So suppose that $B_1(x^i, r_i) \cap C \neq \emptyset$. By definition, $B_1(x^i, r_i)$ is the set of points $y \in \mathbb{R}^n$ such that $\sum_{j=1}^{n} |x^i_j - y_j| \leq r_i$. Hence $B_1(x^i, r_i)$ is defined by

the intersection of $2^n$ halfspaces:

$$B_1(x^i, r_i) = \left\{ y \in \mathbb{R}^n \ \bigg| \ \sum_{j=1}^n a_j(x_j^i - y_j) \leq r_i \ \forall a \in \{-1, +1\}^n \right\}$$

If the intersection of the boundary of $B_1(x^i, r_i)$ with $C$ is one of these hyperplanes, then we are finished. Otherwise, there are at least two of these hyperplanes $H_1 = (a_1, \ldots a_n)$ and $H_2 = (a_1', \ldots, a_n')$ such that $B_1(x^i, r_i) \cap C$ contains a point $y \in H_1 \setminus H_2$ and a point $y' \in H_2 \setminus H_1$, both of which are also on the boundary of $B_1(x^i, r_i)$. Let $j \in [n]$ such that $a_j = -a_j'$. Then $y_j - x_j^i$ has a different sign than $y_j' - x_j^i$, since the fact that $y$ and $y'$ are on the boundary of $B_1(x^i, r_i)$ but on different facets implies that $x_j^i - y_j$ has sign $a_j$ while $x_j^i - y_j'$ has sign $a_j'$. But this contradicts the definition of $C$, since it means that $y$ and $y'$ are on different sides of $Q_j^i$ and hence not in the same cell. $\qquad\square$

This lemma allows us to analyze the number of regions that intersect any cell.

**Lemma 21.** *Let $C$ be a cell. The number of regions that intersect $C$ is at most $2^{O(n)}k^n$.*

*Proof.* For every $S \subseteq P$, the region $r(S)$ is the set of points that are in $B_1(x^i, r_i)$ for all $x_i \in S$ and are not in $B_1(x^i, r_i)$ for all $x_i \notin S$. By Lemma 20, $r(S) \cap C$ is the intersection of $C$ with $k$ halfspaces (one for each $x^i \in P$). It is well-knownthat $k$ halfspaces can divide $\mathbb{R}^n$ into at most $\sum_{i=0}^n \binom{k}{i} = O(n)k^n$ regions, and hence the same bound holds for $C$. $\qquad\square$

Now some standard calculations imply Theorem 19, and hence Theorem 14.

*Proof of Theorem 19.* Lemma 21, together with the fact that there are at most $(k+1)^n$ cells, implies that the number of nonempty regions $m$ is at most $O(n)k^n \cdot (k+1)^n \leq O(n)(k+1)^{2n}$. Since $m = 2^k$, this implies that $2^k \leq O(n)(k+1)^{2n}$. Taking logarithms of both sides yields that

$$k \leq \log(k+1) \cdot O(n), \tag{3}$$

and then taking another logarithm and rearranging yields that $\log n \geq \Omega(\log k - \log \log k) = \Omega(\log k)$ and hence $\log(k+1) \leq O(\log n)$. Plugging this into (3) implies Theorem 19. $\qquad\square$

### 3.3.2 Bounding the Range

In this section we verify the condition for Theorem 16 that every function in $\mathcal{H}$ has its range in $[0, H]$ for $H = O(nC)$. This is actually not quite true as defined, but it is easy enough to ensure: we just consider a restricted class of functions $\mathcal{H}' = \{g_y \mid g_y \in \mathcal{H}, y \in [-C, C]^V\}$. Note that for any fixed set of costs $c$ the class $\mathcal{H}'$ contains $y^*(c)$, so without loss of generality we can just use $\mathcal{H}'$ instead of $\mathcal{H}$. From the definition of pseudo-dimension and $\mathcal{H}' \subseteq \mathcal{H}$, it immediately follows that the pseudo-dimension of $\mathcal{H}'$ is at most that of $\mathcal{H}$. Thus, we just need to ensure that the range of the restricted functions are bounded.

**Lemma 22.** *Each function $g_y \in \mathcal{H}'$ has its range in $[0, H]$ for $H = O(nC)$.*

*Proof.* Let's bound the range by considering the maximum value $g_y$ can take on a set of costs $c$. Recall that $g_y(c) = \|y - y^*(c)\|_1$. Each coordinate can contribute at most $O(C)$ to the sum since $y_i^*(c) \in [-C, C]$ and $y_i \in [-C, C]$. Summing over the $n$ coordinates gives $H = O(nC)$. $\qquad\square$

### 3.3.3 Minimizing the Empirical Loss

Now we give an algorithm to minimize the empirical loss on a collection of sample instances. Let $c_1, c_2, \ldots, c_s$ be a collection of samples from $\mathcal{D}$. Our goal is to find dual prices $y$ minimizing $\frac{1}{s} \sum_{i=1}^s g_y(c_i) = \frac{1}{s} \sum_{i=1}^s \|y - y^*(c_i)\|_1$. Let $x^i = y^*(c_i)$. Then the problem amounts to minimizing $\frac{1}{s} \sum_{i=1}^s \|y - x^i\|_1$ over $y \in [-C, C]^V$. Then, for each coordinate $j$ it suffices to find $y_j$ minimizing $\sum_{i=1}^s \|y_j - x_j^i\|_1$, where $y_j$ and $x_j^i$ denote the $j$-th coordinate of $y$ and $x_i$, respectively. Further, it is easy to see that $\sum_{i=1}^s \|y_j - x_j^i\|_1$ is a continuous piece-wise linear function in $y_j$ where the slope can

| Dataset | Blog Feedback [12] | Covertype | KDD | Skin [10] | Shuttle |
|---|---|---|---|---|---|
| # of Points ($n$) | 52,397 | 581,012 | 98,942 | 100,000 | 43500 |
| # of Features ($d$) | 281 | 54 | 38 | 4 | 10 |

Table 1: Datasets used in experiments based on Euclidean data

change only at $\{x_j^i\}_{i \in [s]}$. Recalling that we can assume wlog that $x^i = y^*(c_i)$ is an integer vector, we only need to consider setting $y_j$ to each value in $\{x_j^i\}_{i \in [s]}$, which is a set of integers. This leads to the following result.

**Theorem 23.** *Given $s$ samples $c_1, c_2, \ldots, c_s$, there exists a polynomial time algorithm which finds integer dual prices $y$ minimizing $\frac{1}{s} \sum_{i=1}^{s} \|y - y^*(c_i)\|_1$.*

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

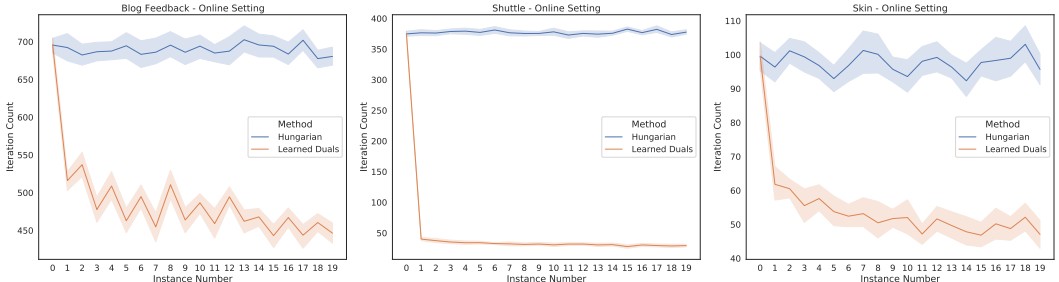

Figure 3: More iteration count results for the Online setting. From left to right, we have the results for the clustering based instances on Blog Feedback, Shuttle, and Skin, all with $k = 500$.

## 5  Extending to $b$-Matching

We now extend the results from Section 3 to the minimum weight perfect $b$-matching problem on bipartite graphs. In the extension we are given a bipartite graph $G = (V, E)$, where $V = L \cup R$, a weight vector $c \in \mathbb{Z}_+^E$ and a demand vector $b \in \mathbb{Z}_+^V$. As before, we assume that the primal is feasible for the remainder of this section. Note that the feasibility of the primal can be checked with a single call to a maximum flow algorithm.

The problem is modeled by the following linear program and its dual linear program.

$$
\begin{aligned}
\min \quad & \sum_{e \in E} c_e x_e \\
& \sum_{e \in \delta(i)} x_e = b_i \quad \forall i \in V \\
& x_e \geq 0 \qquad \forall e \in E
\end{aligned}
\tag{MWBM-P}
$$

$$
\begin{aligned}
\max \quad & \sum_{i \in V} b_i y_i \\
& y_i + y_j \leq c_{ij} \quad \forall ij \in E
\end{aligned}
\tag{MWBM-D}
$$

First we show how to project an infeasible dual onto the set of feasible solutions, then we give a simple primal dual scheme for moving to an optimal solution. The end goal of this section is proving the following theorem.

**Theorem 24.** *There exists an algorithm which takes as input a (not necessarily feasible) dual assignment $y$ and finds a minimum weight perfect $b$-matching in $O(mn\|y^* - y\|_1)$ time, where $y^*$ is an optimal dual solution and $\|y^* - y\|_{b,1} := \sum_i b_i |y_i^* - y_i|$.*

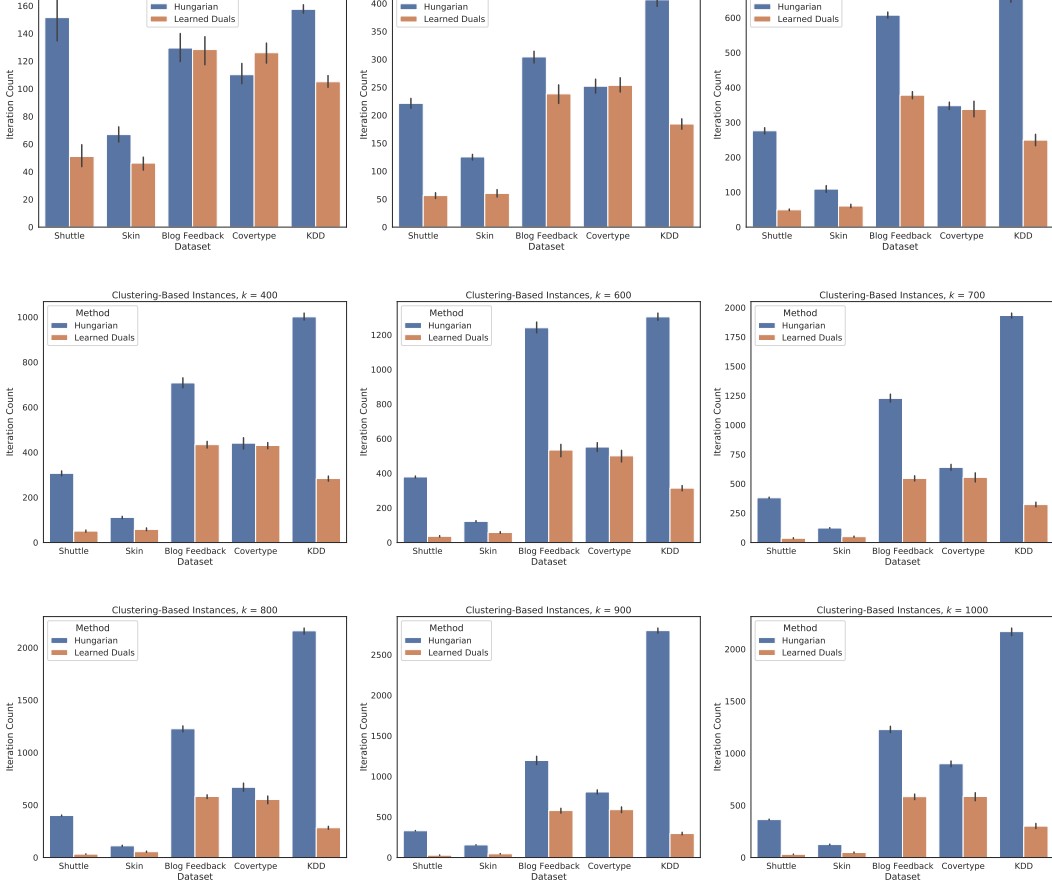

Figure 4: Iteration count results for clustering derived instances in the Batch setting on other values of $k$. Here we give the results for each $k$ in $\{100 \cdot i \mid 1 \le i \le 10\} \setminus \{500\}$.

## 5.1 Recovering a Feasible Dual Solution for $b$-Matching

As in Section 3, our goal now is to find non-negative perturbations $\delta$ such that $\hat{y}' := \hat{y} - \delta$ is feasible for (MWPM-D). We would like these perturbations to preserve as much of the dual objective value as possible. Again we define $r_e := \hat{y}_i + \hat{y}_j - c_e$ for each edge $e = ij \in E$. Following the same steps as before, this leads to the following linear program and it's dual.

$$\begin{aligned}
\min \quad & \sum_{i \in V} b_i \delta_i \\
& \delta_i + \delta_j \ge r_e \quad \forall e = ij \in E \\
& \delta_i \ge 0 \qquad \forall i \in V
\end{aligned} \tag{4}$$

$$\begin{aligned}
\max \quad & \sum_{e \in E} r_e \gamma_e \\
& \sum_{e \in N(i)} \gamma_e \le b_i \quad \forall i \in V \\
& \gamma_e \ge 0 \qquad \forall e \in E
\end{aligned} \tag{5}$$

Again we are interested in finding a fast approximate solution to this problem. We develop a new algorithm different than that used in the prior section and show it is a 2 approximation to (4). To do so, consider the dual LP above. This is an instance of the weighted $b$-matching problem where edges can be selected any number of times. We will first develop a 2 approximation to this LP in

$O(m \log m + n)$ time. The analysis will be done via a dual fitting analysis. This analysis will give us the corresponding 2-approximate fractional primal solution that will be used to construct $\hat{y}'$.

Consider the following algorithm for the dual problem. Sort the edges $e$ in decreasing order of $r_e$. When considering an edge $e' = i'j'$ in this order set $\gamma_{e'}$ as large as possible such that $\sum_{e \in N(i')} \gamma_e \leq b_{i'}$ and $\sum_{e \in N(j')} \gamma_e \leq b_{j'}$. Notice the running time of the algorithm is bounded by $O(m \log m + n)$.

When the algorithm terminates we construct a corresponding primal solutions. For each $i \in V$, set $\delta_i = \frac{\sum_{e \in N(i)} \gamma_e r_e}{b_i}$. That is, $\delta_i$ is the summation of the weights $r$ of the adjacent edges divided by the $b$-matching constraint value $b_i$. We will show that $\delta$ is a feasible primal solution. Moreover that the primal and dual objectives are within a factor two of each other.

**Lemma 25.** *The solution $\delta$ is feasible for LP (4) and $\gamma$ is feasible for the dual LP (5).*

*Proof.* The feasibility for the dual is by construction, so consider the primal. Consider any edge $e' = i'j'$. Our goal is to show that $\delta_{i'} + \delta_{j'} \geq r_{e'}$. Let $A_{e'}$ be the set of edges considered by the algorithm up to edge $e'$ including the edge itself. These edges have weight at least as large $e'$. We claim that either $\sum_{e \in N(i') \cap A_{e'}} \gamma_e = b_{i'}$ or $\sum_{e \in N(j') \cap A_{e'}} \gamma_e = b_{j'}$. Indeed, otherwise we would increase $\gamma_{e'}$ until this is true. Without loss of generality say that $\sum_{e \in N(i') \cap A_{e'}} \gamma_e = b_{i'}$. We will argue that $\delta_{i'} \geq r_{e'}$. Knowing that $\delta_{j'}$ is non-negative, this will complete the proof.

Consider the value of $\delta_{i'}$. This is $\frac{\sum_{e \in N(i')} \gamma_e r_e}{b_{i'}} = \frac{\sum_{e \in N(i') \cap A_{e'}} \gamma_e r_e}{b_{i'}}$. We know from the above that $\sum_{e \in N(i') \cap A_{e'}} \gamma_e = b_{i'}$ and every edge in $A_{e'}$ has weight greater than $e'$. Thus, $\frac{\sum_{e \in N(i') \cap A_{e'}} \gamma_e r_e}{b_{i'}} \geq r_{e'} \frac{\sum_{e \in N(i') \cap A_{e'}} \gamma_e}{b_{i'}} = r_{e'}$ $\square$

Next we bound the objective of the primal as a function of the dual.

**Lemma 26.** *The primal objective is exactly twice the dual objective.*

*Proof.* It suffices to show each edge $e = ij$ contributes twice as much to the primal objective as it does to the dual objective. First, $e$'s contribution to the dual objective is clearly $r_e \gamma_e$. For the dual, edge $e$ contributes to the summation for both end points. That is, $e$ contributes to $\delta_i$ by $\gamma_e r_e / b_i$ and to $\delta_j$ by $\gamma_e r_e / b_j$. Thus, edge $e$'s contribution to the primal objective is $b_i \frac{\gamma_e r_e}{b_i} + b_j \frac{\gamma_e r_e}{b_j} = 2 \gamma_e r_e$, as desired. $\square$

Thus, we have found a 2-approximate solution to the primal LP (4). However, the solution is not necessarily integral. Thus, to make it integral, we do the following simple rounding:

$$\delta_i \leftarrow \begin{cases} \lfloor 2\delta_i \rfloor & \text{if } \delta_i \geq 0.5 \\ 0 & \text{if } \delta_i \in [0, 0.5) \end{cases}$$

Clearly this update can double the cost in the worst case. Hence we only need to check that every constraint remains satisfied. To see this consider an edge $e = ij$ and let $\delta_i$ and $\delta_j$ be the dual values before the update. Note that $r_e$ is an integer assuming that we are given integer dual values $\hat{y}$. Assume $r_e \geq 1$ since otherwise the constraint trivially holds true. It is an easy exercise to see that $\lfloor 2x \rfloor \geq x$ for all $x \geq 0.5$. Thus, if $\delta_i, \delta_j \geq 0.5$, then the update only increases the value of $\delta_i$ and $\delta_j$, keeping the constraint satisfied. Further, as $r_e \geq 1$, it must be the case that $\delta_i \geq 0.5$ or $\delta_j \geq 0.5$. So, we only need to consider the case either $\delta_i \geq 0.5$ and $\delta_j < 0.5$; or $\delta_i < 0.5$ and $\delta_j \geq 0.5$. Assume wlog that the latter is the case. Since $\delta_i \leq \delta_j$, if $\delta_i + \delta_j \geq r_e$, we have $2\delta_j \geq r_e$. Then, we have $\lfloor 2\delta_j \rfloor \geq r_e$ as $r_e$ is an integer. Again, the constraint is satisfied.

Thus, we obtain the following which is analogous to Theorem 7.

**Theorem 27.** *There is a $O(m \log m + n)$ time algorithm that takes an infeasible integer dual $\hat{y}$ and constructs a feasible integer dual $\hat{y}'$ such that $\|\hat{y} - \hat{y}'\|_{b,1} \leq 4\|y^* - \hat{y}\|_{b,1}$ where $y^*$ is the optimal dual solution. Thus, we have $\|\hat{y}' - y^*\|_{b,1} \leq 5\|\hat{y} - y^*\|_{b,1}$.*

## 5.2 Converting a Feasible Dual Solution to an Optimal Primal Solution

Now we consider taking a feasible dual $y$ and moving to an optimal solution for the $b$-matching problem. The algorithm we use is a simple primal-dual scheme that generalizes Algorithm 2. See Algorithm 3 for details. Below we give a brief analysis of this algorithm. The objective is to establish a running time in terms of the following distance $\|y^* - y\|_{b,1} := \sum_i b_i |y_i^* - y_i|$. One can view this distance as the $\ell_1$ norm distance where each coordinate axis is given a different level of importance by the $b_i$ values.

---

**Algorithm 3** Simple Primal-Dual Scheme for MWBM

---

1: **procedure** MWBM-PRIMALDUAL($G = (V, E), c, y$)
2:   $E' \leftarrow \{ij \in E \mid y_i + y_j = c_{ij}\}$          ▷ Set of tight edges in the dual
3:   $G' \leftarrow (L \cup R \cup \{s, t\}, E' \cup \{si \mid i \in L\} \cup \{jt \mid j \in R\})$      ▷ Network of tight edges
4:   $\forall e \in E(G')$ s.t. $e = si$ or $e = it$, $u_e \leftarrow b_i$
5:   $u_e \leftarrow \infty$ for all other edges of $G'$
6:   $f \leftarrow$ Maximum $s - t$ flow in $G'$ with capacities $u$
7:   **while** Value of $f$ is $< \sum_{i \in L} b_i$ **do**
8:     Find a set $S \subseteq L$ such that $\sum_{i \in S} b_i > \sum_{j \in \Gamma(S)} b_j$      ▷ Exists by Lemma 28
                                                         ▷ Can be found in $O(m + n)$ time
9:     $\epsilon \leftarrow \min_{i \in S, j \in R \setminus \Gamma(S)} \{c_{ij} - y_i - y_j\}$
10:    $\forall i \in S, y_i \leftarrow y_i + \epsilon$
11:    $\forall j \in \Gamma(S), y_j \leftarrow y_j - \epsilon$
12:    Update $E', G', u$
13:    $f \leftarrow$ Maximum $s - t$ flow in $G'$ with capacities $u$
14:   $x \leftarrow f$ restricted to edges of $G$
15:   Return $x$

---

First we consider the correctness of the algorithm. As before, we need to show that the update rule is well defined. The following is a well known generalization of Hall's theorem, showing that line 8 is well defined. Further, the step can be implemented efficiently given $f$. The proof closely follows that of Proposition 8 – the only difference is factoring $b$ in the matching size and vertex cover size.

**Proposition 28.** *Let $G'$ be the flow network defined in Algorithm 3 with capacities $\rho$ and let $f$ be the maximum $s - t$ flow in $G'$ if the value of $f$ is less than $\sum_{i \in L} b_i$ then there exists $S \subseteq L$ such that $\sum_{i \in S} b_i > \sum_{j \in \Gamma(S)} b_j$. Further, such $S$ can be found in $O(m + n)$ time.*

The following is analogous to Proposition 9 in Section 3.

**Proposition 29.** *Let $y$ be dual feasible and suppose that $S \subseteq L$ with $\sum_{i \in S} b_i > \sum_{j \in \Gamma(S)} b_j$ in $G'$. Let $\epsilon = \min_{i \in S, j \in R \setminus \Gamma(S)} \{c_{ij} - y_i - y_j\}$. Then as long as $c$ and $y$ are integers we have $\epsilon \geq 1$.*

Additionally, we need to establish that $y$ remains feasible throughout the execution of the algorithm. This is nearly identical to the corresponding lemma in Section 3 so we state it as the following lemma without proof.

**Lemma 30.** *If Algorithm 3 is given an initial dual feasible $y$, then $y$ remains dual feasible throughout its execution.*

The above statements can be combined to give the following theorem.

**Theorem 31.** *There exists an algorithm for minimum weight perfect $b$-matching in bipartite graphs which runs in time $O(nm\|y^* - y\|_{b,1})$, where $y^*$ is an optimal dual solution and $y$ is the initial dual feasible solution passed to the algorithm.*

*Proof.* The correctness of the algorithm is implied by Lemma 30 and the fact that the flow network $G'$ ensures that the resulting solution $x$ that it finds satisfies complementary slackness with $y$. Thus we just need to establish the running time.

Note that it suffices to bound the number of iterations in terms of $O(\|y^* - y\|_{b,1})$ since the most costly step of each iteration is finding the maximum flow in the network $G'$, which can be done in time $O(nm)$. The two propositions above state that the net increase in the dual objective is always at least 1, and so the number of iterations is at most $\sum_i b_i y_i^* - \sum_i b_i y_i \leq \sum_i b_i \|y_i^* - y_i\| = \|y^* - y\|_{b,1}$. □

This theorem, combined with Theorem 27, gives Theorem 24, as desired.

## 5.3 Learning the Dual Prices

In this section we extend the results from Section 3.3 to the case of $b$-matching. As before, we consider a graph with fixed demands $b$ and an unknown distribution $\mathcal{D}$ over the edge costs $c$. We are interested in learning a fixed set of prices $y$ which is in some sense best for this distribution. Since the running time of the algorithms we consider depends on $\|y^* - y\|_{b,1}$ it is natural to choose this as our loss function with respect to the learning task. Thus we define $L_b(y, c) = \|y - y^*(c)\|_{b,1}$, where again $y^*(c)$ is a fixed optimal dual vector for costs $c$. Our goal is to perform well against the best choice for the distribution. Formally, let $y^* := \arg\min_y \mathbb{E}_{c \sim \mathcal{D}}[L_b(y, c)]$. Additionally, let $C$ be a bound on the edge costs and $B = \max_{i \in V} b_i$ be a bound on the demands. We have the following result which is analogous to Theorem 14.

**Theorem 32.** *There is an algorithm that after* $s = O\left(\left(\frac{nCB}{\epsilon}\right)^2 (n \log n + \log(1/\rho))\right)$ *samples returns integer dual values* $\hat{y}$ *such that* $\mathbb{E}_{c \sim \mathcal{D}}[L_b(\hat{y}, c)] \leq \mathbb{E}_{c \sim \mathcal{D}}[L(y^*, c] + \epsilon$ *with probability at least* $1 - \rho$. *The algorithm runs in time polynomial in* $n, m$ *and* $s$.

At a high level, we can prove this theorem by again applying Theorem 16 and Corollary 17. To do this we define the following family of functions $\mathcal{H}_b = \{g_y \mid y \in \mathbb{R}^V\}$ where $g_y = \|y - y^*(c)\|_{b,1}$. We need to verify the following: (1) the range of these functions are bounded in $[0, H]$ for some $H = O(nCB)$, (2) minimizing the empirical loss can be done efficiently, and (3) the pseudo-dimension of $\mathcal{H}_b$ is bounded by $O(n \log n)$. Applying similar arguments as in Sections 3.3.2 and 3.3.3 give us the first two points. Here we focus on the last point, bounding the pseudo-dimension.

Note that for $b \in \mathbb{R}^n_+$, $\|\cdot\|_{b,1}$ is a norm. Intuitively, the geometry induced by $\|\cdot\|_{b,1}$ is the same as the geometry induced by $\|\cdot\|_1$ except some axes are stretched by an appropriate amount. This should imply that the functions in $\mathcal{H}_b$ should not be more complicated than the functions in $\mathcal{H}$. We make this intuition more formal by arguing that we can map from one setting to the other while preserving membership in the respective balls induced by these norms. The following key lemma will imply that the pseudo-dimension of $\mathcal{H}_b$ is no larger than the pseudo-dimension of $\mathcal{H}$.

**Lemma 33.** *Let* $B_{b,1}(x, r) = \{y \mid \|x - y\|_{b,1} \leq r\}$ *and* $B_1(x, r) = \{y \mid \|x - y\|_1 \leq r\}$ *be the balls of radius* $r$ *under each norm, respectively. There is a mapping* $\phi : \mathbb{R}^n \to \mathbb{R}^n$ *such that* $y \in B_{b,1}(x, r)$ *if and only if* $\phi(y) \in B_1(\phi(x), r)$.

*Proof.* Define $\phi(y)_i = b_i y_i$ for $i = 1, 2, \ldots, n$. Now we have the following which implies the lemma.

$$\|x - y\|_{b,1} = \sum_i b_i |x_i - y_i| = \sum_i |b_i x_i - b_i y_i|$$
$$= \|\phi(x) - \phi(y)\|_1$$

Thus one of these is at most $r$ if and only if the other is. □

Now define the family of functions $\mathcal{H}_{b,n} = \{f_y : \mathbb{R}^n \to \mathbb{R} \mid y \in \mathbb{R}^n, f_y(x) = \|y - x\|_{b,1}\}$, we have the following which is analogous to Lemma 18.

**Lemma 34.** *The pseudo-dimension of* $\mathcal{H}_b$ *is at most the pseudo-dimension of* $\mathcal{H}_{b,n}$

*Proof.* Nearly identical to that of Lemma 18 but with $\|\cdot\|_1$ replaced with $\|\cdot\|_{b,1}$. □

We can now prove that the pseudo-dimension of $\mathcal{H}_b$ is bounded by $O(n \log n)$.

**Lemma 35.** *The pseudo-dimension of* $\mathcal{H}_b$ *is at most* $O(n \log n)$.

*Proof.* By Lemma 34 we have that the pseudo-dimension of $\mathcal{H}_b$ is at most $\mathcal{H}_{b,n}$. We now show that the pseudo-dimension of $\mathcal{H}_{b,n}$ is at most the pseudo-dimension of $\mathcal{H}_n$ using Lemma 33. Let $x^1, \ldots, x^k \in \mathbb{R}^n$ be given. Now consider $y^j = \phi(x^j)$ for $j = 1, \ldots, k$. By Lemma 33 we can see that $x^1, \ldots, x^k$ are shattered by $\mathcal{H}_{b,n}$ if and only if $y^1, \ldots, y^k$ are shattered by $\mathcal{H}_n$. Thus the pseudo-dimension of $\mathcal{H}_{b,n}$ is at most $\mathcal{H}_n$ and then the lemma follows by Theorem 19. □

# 6  Conclusion and Future Work

In this work we showed how to use learned predictions to warm-start primal-dual algorithms for weighted matching problems to improve their running times. We identified three key challenges of feasibility, learnability and optimization, for any such scheme, and showed that by working in the dual space we could give rigorous performance guarantees for each. Finally, we showed that our proposed methods are not only simpler, but also more efficient in practice.

An immediate avenue for future work is to extend these results to other combinatorial optimization problems. The key ingredient is identifying an appropriate intermediate representation: it must be simple enough to be learnable with small sample complexity, yet sophisticated enough to capture the underlying structure of the problem at hand.

## Acknowledgments and Disclosure of Funding

Michael Dinitz was supported in part by NSF grant CCF-1909111. Sungjin Im was supported in part by NSF grants CCF-1617653 and CCF-1844939. Thomas Lavastida and Benjamin Moseley were supported in part by NSF grants CCF-1824303, CCF-1845146, CCF-1733873 and CMMI-1938909. Benjamin Moseley was additionally supported in part by a Google Research Award, an Infor Research Award, and a Carnegie Bosch Junior Faculty Chair.

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

## A  Additional Experimental Results

Here we present additional experimental results that were omitted from Section 4. First we present our results while looking at the running time as opposed to the number of primal dual iterations.

## A.1 Running Time

Figure 5 gives running time results for the batch setting, while Figure 6 give the results for the online setting. Finally, Figure 7 looks at the clustering derived instances for other values of $k$. We see similar performance improvements for Learned Duals against the standard Hungarian algorithm, showing that the impact of running Algorithm 1 to make the predicted duals feasible is minimal.

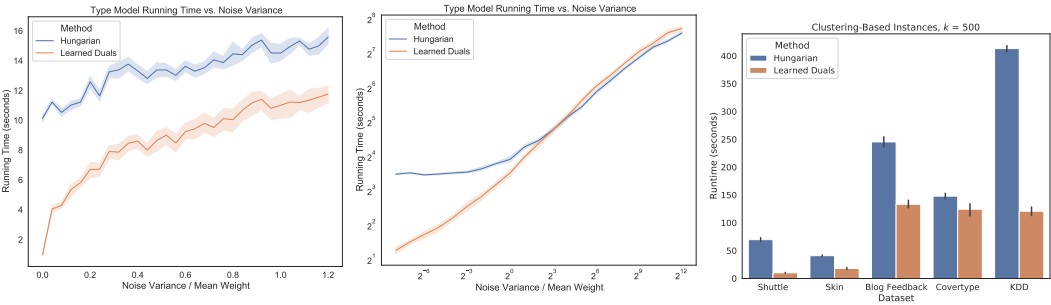

Figure 5: Running time results (in seconds) for the Batch setting. The left figure gives the iteration count for the type model (synthetic data) versus linearly increasing $v$, while the middle geometrically increases $v$. The right figure summarizes the results for clustering based instances (real data) in the batch setting.

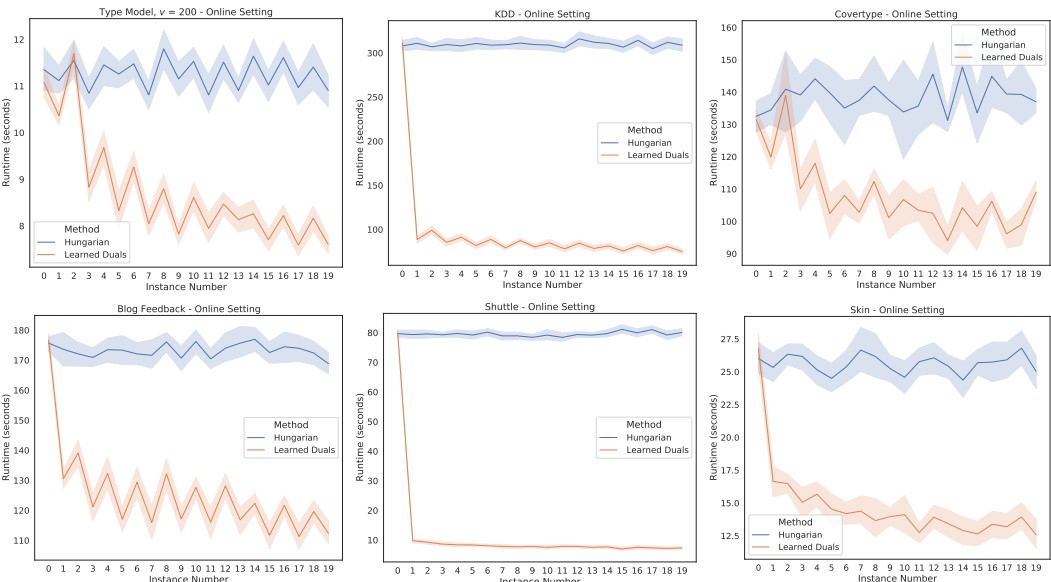

Figure 6: Running time results for the Online setting. The top left figure is for the type model (synthetic data). The rest, in order, are KDD and Covertype, Blog Feedback, Shuttle, and Skin. All use $k = 500$.

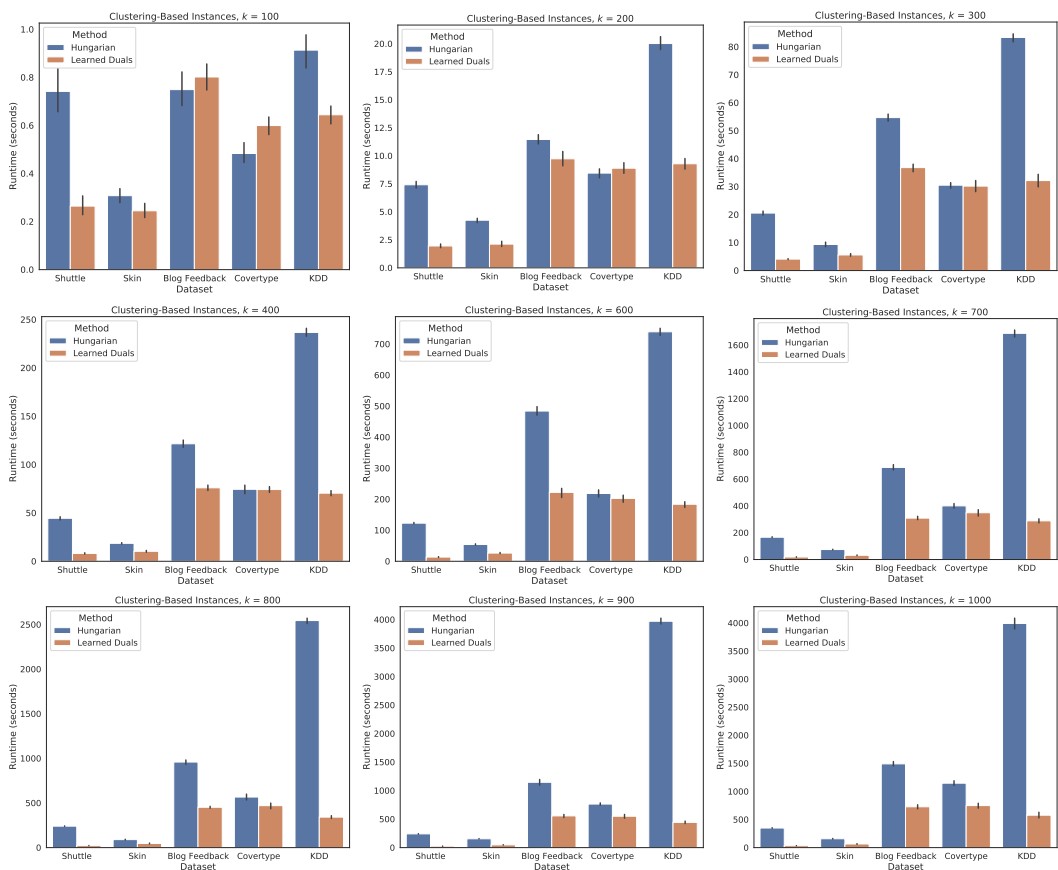

Figure 7: Running time results (in seconds) for clustering derived instances in the Batch setting on other values of $k$. Here we give the results for each $k$ in $\{100 \cdot i \mid 1 \leq i \leq 10\} \setminus \{500\}$.