# OpenReview forum: "Faster Matchings via Learned Duals"
_NeurIPS.cc/2021/Conference — NeurIPS 2021 Oral_

### Official Review · Reviewer_RBoF · 2021-07-09

**Rating:** 7
**Confidence:** 4

**Summary:**

In this paper the authors propose a learning algorithm for accelerating solving of bipartite minimum weight perfect matching problems, which are combinatorial optimization problems that can be solved in polynomial time. These problems can be represented as linear programs, and they propose to learn the initial point to the dual problem, which, when feasible, can be fed to the Hungarian algorithm to kick-start the optimization process. Given a collection of training and testing instances, they propose to solve the training instance to obtain their optimal dual solutions and take their median as a learned initial dual point. Given a new test instance, they propose to take this training median as initial point, make it feasible for the test instance using a greedy heuristic, and solve the test instance using the Hungarian algorithm initialized with this feasible dual point. They compute (Theorem 1) the sample and time complexity of the algorithm, and show empirical improvements on two distribution of problems compared to a vanilla Hungarian algorithm.

**Limitations And Societal Impact:**

The approach is developed for a specific problem (bipartite minimum weight perfect matching), which is used in ad allocation. The authors make this clear. The lack of experiments on very large graphs makes it difficult to know how the gains would scale up, although I would expect at least some gains on larger graphs. I am not concerned about adverse negative societal impact of this work.

**Main Review:**

I think the basic content of the paper is okay. In terms of philosophy, the paper is initially presented as being very general: their goal is to speed up primal-dual algorithms by learning initial dual values, and they present their work as being an initial step in that direction. The larger class of problems they want to help improve is not very clear, but I assume they would be convex problems that are commonly solved using interior point methods?

I think the goal is fine (this is certainly a way among many others machine learning could be helpful in improving mathematical optimization), but I don't really know if there is a way to generalize so easily the results of the paper towards that goal: mostly, their solution to the first problem (feasibility) seems very specific to the problem at hand, and I wonder whether it would be possible at all to design reasonable, and fast, heuristics for arbitrary convex problems. That is, I feel their challenge 1 (Feasibility) would be quite difficult, and that it is really not clear whether it can be solved in general. (I'd be happy to hear if the authors have any ideas in this direction, however.)

Disregarding the way the paper is sold, and focusing on the content, I think the paper makes a sensible contribution. The theory is the main contribution here (Theorem 3). Considering how NeurIPS is a machine learning conference, here is something a bit primitive about using the median of the dual solutions as initial point, instead of having, say a real machine learning model trained on the dual solutions, with features characterizing the weights, that would be used at test time. But I suppose this would make the theory harder, and since this is, as far as I know, the first work of this kind, this is not a strong objection.

In terms of experiments, I find the "instance generation from real instances" somewhat bizarre. I feel the authors tried to have graphs coming somehow from real data, but the procedure seems to be completely artificial. It would have been substantially better to have real graphs from real problems (for example, the ad allocation applications that are mentioned in Section 1.2.) Right now the problems seem to me still synthetic. I can't really say they have demonstrated empirical gains on the kinds of instances that one would actually encounter in real life. The size of the training and test sets are also seem quite small: 20 and 10 instances. But overall, I don't think these criticisms are enough to make me reject the paper.

In terms of structure, I thought the paper was clear, if perhaps a bit repetitive with the informal/formal duplication. I think the fact that they use the median of the dual points was quite buried in the text, and it should have been much more clear: as a machine learning reader, my first instinct was so ask, if you a learning to predict an initial dual point, what is the model and what are the features? This should have been in the abstract and the introduction but it was quite hidden as almost unimportant. I would like to see this improved if possible.

Overall, my recommendation would be acceptance of the paper.

---
Edit after rebuttal

Thank you for your interesting answers to my questions. I think you made good points regarding near-optimality of the median that I had not thought about during my review. I increased my score to "Accept".

**Time Spent Reviewing:**

5

---

> ### Author Response · Authors · 2021-08-09
> **Thank you for your thorough review and suggestions**
>
> Thank you for your thorough review and suggestions. We particularly appreciate your suggestion to make the learning model clear in the beginning.
>
> For the question of what other problems this can approach generalize to, we believe that the larger class of primal-dual, “potential-based” algorithms could be a ripe avenue for future work. These algorithms proceed by iteratively improving a potential function, and what we show here is that starting with a learned near-optimal solution (as opposed to a naive all-zeros solution) leads to improvements both in theory and in practice. Of course, the challenge, as the reviewer insightfully points out, is in the feasibility projection step. Here we show a problem specific version, but perhaps future researchers can find more general techniques. Overall, we view our work as the first to use learning to give faster algorithms, both theoretically and empirically, tackling one of the most basic combinatorial optimization problems.
>
> We agree with the reviewer that using the median (as the empirical risk minimizer) is a particularly simple learning algorithm, but we view this as a strength rather than a weakness: even very naive learning algorithms can be plugged into our framework with great success!  More elaborate learning algorithms, including those that take graph features into account, could be plugged into our framework without change (simply replacing the learning algorithm and feeding the learned duals into the rest of the framework), and if it happens to give better duals, then we will automatically have improved performance.  But we note that as we prove in Section 3.3, the median (as ERM) gives essentially optimal duals after only a polynomial number of samples.  Of course, there could be a gap between theory and practice -- a more complicated ML algorithm could learn better duals, or learn them faster.  So we really view the median (ERM) algorithm as a “proof of concept” that the duals can actually be learned reasonably.  This is important since much of the prior work on “algorithms with learned advice” proposes using advice that helps the algorithm, but is not obviously learnable.  We need to make sure that the duals are actually learnable!
>
> Finding a public dataset with the exact desiderata we need has turned out to be a challenge, which is why we resorted to modifying non-matching instances to fit the mold we consider here. The few advertising datasets that are public are typically geared towards improving click prediction models, or to understand bidding behavior, and so do not have the temporal and structural attributes that we need.  Hence we chose to create matching instances based on real, publically available geometric data. While we agree with the reviewer that the empirical results are semi-synthetic due to the way repeated graphs are generated, we still see this as a strong indicator for our approach’s practicality and effectiveness.

---

### Official Review · Reviewer_pG4Y · 2021-07-16

**Rating:** 7
**Confidence:** 3

**Summary:**

any combinatorial optimization problems may be solved or approximated with primal-dual algorithms built on a linear programming relaxation of the problem. These algorithms maintain an infeasible primal solution that is driven towards feasibility and a (matching) feasible dual solution that is driven towards optimality. The algorithm terminates once the primal solution becomes feasible and uses the feasible dual as a quality guarantee in the analysis.

The aim of the submission is to improve the running time of one particular primal-dual algorithm -- the Hungarian method for weighted bipartite matchings -- by using machine learning. The goal is to augment the algorithm with learned predictions that improve the (theoretical and empircal) running time if good, and that fall back to the theoretical worst-case otherwise.

Instead of trying to predict a primal solution (something the literature has determined to be difficult for bipartite matchings), the authors propose to predict a dual solution that is already driven some way towards optimality and to use that solution to "warm-start" the algorithm. The difficulty here is that the prediction might not be a feasible dual solution (and thus useless for the Hungarian method). The authors fix this issue by providing a (linear time) procedure that turns an infeasible predicted dual into a feasible dual while increasing the distance to the optimum dual by at most a factor of 3. Finally, the authors prove that duals can be learned approximately from O~(C^2n^3) samples if the matching instances are drawn from a fixed, but unknown distribution (C being the maximum weight of any edge).

To argue that the learning provides an empirical improvement, the authors compare their warm-started method against the classical Hungarian method. The results convinced me that the warm-started method works well.

**Limitations And Societal Impact:**

If practicability is a big concern, a comparison with min-cost flow algorithms would add value to the contribution.

**Main Review:**

The paper is very well written and well structured. The contribution extends an interesting line of theoretical research that works along similar lines and I would be happy to see develop a field here that systematically works through classical algorithms. This contribution would make good first steps. The research is certainly original and I find the goal of having a practical algorithm is achieved here; the implementation overhead to include learned duals seems minimal.

If we care deeply about practicability, there could be alternatives to the Hungarian method: Weighted bipartite matchings may be solved as minimum cost flow problems for which we have efficient implementations:
- the double scaling algorithm runs in O(nm log nC) and matches the theoretical worst-case running time of the Hungarian method
- the push relabel method is another primal-dual algorithm that is known to perform very well in practice (but is slower in theory)
- the network simplex algorithm has no comparable guarantees but is known to perform very quickly in practice.

While the authors discuss more efficient algorithms for bipartite matchings (and convincingly dismiss them as impractical), this alternative is not discussed.

However, from a theoretical point of view, studying the Hungarian method makes a lot of sense to me, and it is not clear if minimum cost flows would be a faster alternative in a practical setting. I would therefore see this as a minor concern.

**Time Spent Reviewing:**

3

---

> ### Author Response · Authors · 2021-08-09
> **Thank you for your careful review and suggestions**
>
> Thank you for suggesting alternative points of comparison. We believe that the Hungarian algorithm is the most popular (and fastest) way of solving min-cost bipartite matchings in practice (rather than applying general min-cost flow algorithms), but we agree with the reviewer that it would have been interesting to compare to min-cost flow. Due to the tight connection between the Hungarian algorithm and min-cost flows, we believe that our algorithm will be significantly faster.

---

### Official Review · Reviewer_9XRM · 2021-07-16

**Rating:** 8
**Confidence:** 4

**Summary:**

The paper gives a learning-augmented algorithm to improve the running time for the minimum weight perfect matching (MWPM) problem in bipartite graphs. For a given bipartite graph, they consider the setting that the vector of (integral) edge weights is sampled from a probability distribution. For any given (unknown) distribution, they show that
- one can efficiently learn dual variables that minimize the expected l1-distance to the optimal dual variables for a random instance sampled from the given distribution
- these learned dual variables can be efficiently turned into a *feasible* dual solution for the given instance at the expense of increasing the aforementioned l1-distance by a factor at most 3.
- using this feasible dual solution to warm-start the known Hungarian method algorithm to solve MWPM, the running time improves from O(mn) to O(min{mn,m\sqrt{n}alpha}), where alpha is the aforementioned minimal expected l1-distance for the given distribution.

Additionally, they provide an extensive experimental evaluation confirming an improved running time when the input distribution is sufficiently concentrated, and a running time comparable to the classical Hungarian method algorithm when the distribution is not concentrated. On instances generated from real-world data, the observed running time improvement was of the order of roughly a factor 2.

**Ethical Concerns:**

No ethical concerns.

**Limitations And Societal Impact:**

Limitations are adequately addressed. There is no discussion of potential negative societal impacts, but I also do not think that there are any relevant negative impacts.

**Main Review:**

The most interesting aspect of this paper is that in contrast to most work in the field of learning-augmented algorithms, which uses predictions to improve the competitive ratio, this paper uses predictions to improve the running time instead. A theoretical improvement over the Hungarian method is obtained provided that the input distribution is such that the minimal l1-distance alpha of the best "average" dual solution to the actual dual optimum is less than sqrt{n}. As pointed out in the paper, there also exist other algorithms for MWPM than the Hungarian method, and those algorithms achieve a running time of O(m\sqrt{n}*log(nC)), where C is the maximal edge weight. A theoretical improvement over these algorithms is achieved provided that alpha is less than log(nC). The experiments do not include these latter algorithms, but the authors argue that the Hungarian method is the standard algorithm in practice and easy to implement.

In terms of techniques, the proofs seem relatively simple, but this does not diminish the value of the paper. In particular, I find it interesting and appealing that the learnt part are dual variables. There is a good chance that the methods of this paper extend to additional problems.

A potential area of improvement is to complement the argument from line 77 about the primal solution being brittle (and therefore not useful for predictions) by an argument that the dual solution is less brittle. In other words, some kind of justification that the quantity alpha in Theorem 1 is small if the input distribution is sufficiently concentrated. Otherwise it is not clear in what settings the output of your learning algorithm is useful.

The paper is well-written and easy to understand. In my opinion, it is a significant contribution to the young field of learning-augmented algorithms, and I recommend acceptance.

Minor comments:
- Line 32: Antoniadis et al [3] is not about the secretary problem, but about caching and the more general MTS problem.
- Line 234: A letter has a funny color.
- Line 236-239: Where/What is the reference (3)?
- Algorithm 2: I do not think you defined the notation L and R used here (I understand it is the left/right half of the bipartite graph)
- Line 307: minimizes -> minimize
- Line 319: by -> is defined by

**Time Spent Reviewing:**

3

---

> ### Author Response · Authors · 2021-08-09
> **Thank you for your careful review and comments**
>
> Thank you for your careful review and comments. We will address all of your comments.
> To answer your question “Line 236-239: Where/What is the reference (3)?”, it was supposed to refer to the dual LP (MWPM-D). We apologize for this. It was correctly referring to the dual in the full version which can be found in the supplementary materials; but in the submission version, it was not.
>
> Experimentally, we observed that the optimum matchings significantly varied even under mild perturbation of the input. Formally showing duals are more desirable to learn than primals under perturbation would be very interesting.

---

### Official Review · Reviewer_gyoj · 2021-07-18

**Rating:** 7
**Confidence:** 3

**Summary:**

The paper proposed a learning-augmented algorithm that receives a learned dual solution and uses it as a warm-start to improve upon the running time of the Hungarian method for weighted bipartite matching. The paper provides a neat 3-stage approach for obtaining “faster” algorithms for matching problem: 1) Learn a helpful dual solution for a given instance, 2) Convert a given predicted dual solution to a feasible dual solution, and 3) Lastly, get a near-optimal solution starting from the feasible “warm-start” solution. The paper provides an end-to-end algorithm for this problem using the described approach.

**Limitations And Societal Impact:**

I do not see any potential negative societal impact of their work.

**Main Review:**

Putting all pieces together, the paper gives $\tilde{O}(m \sqrt{n} \cdot \min{\alpha, \sqrt{n}})$ where $\alpha = \min_y \mathbb{E}_{c~D} [||y  – y^*(c)||_1]$ where $D$ is the distribution over the inputs of interest, $c$ is the weight vector of the edges in the underlying graph and $y^*(c)$ is the optimal dual solution of matching w.r.t. edge weights $c$. In particular, if the prediction vector (i.e., $y$) is close enough to the actual dual values, the algorithm shaves a factor of $\sqrt{n}$ from the running time of the Hungarian method.
While the algorithm provides an improvement over the Hungarian method if the predictions are good, it is not clear how it compares to the fastest known algorithm for weighted bipartite matching. Though, given the simplicity and applicability of the Hungarian method the result is still of interest.

Comments:
1.	Theorem 3 does not provide the running time for learning $\hat{y}$
2.	Proof of Theorem 19 (in appendix) is hard to follow. Line 452: $2^d$ or $2^n$? Line 453: y_i or x_i? Why is it clear that the number of hyperplane is $O(kn)$?
3.	Paragraph starting at line 346 in the experiment. The description is a bit unclear. What is $(i,j)$?
4.	While the results are believable the proofs get informal throughout the paper. It would be beneficial if authors work further on the presentation of the technical part of the paper (specifically section 3.3.1).
5.	Fix all instance of double quotation. For example, “warm-start” in the abstract and line 42, “solution” in line 42
Overall, the paper provides an interesting approach in the design of algorithms with prediction. To best of my knowledge it is the first time these techniques have been used to improve the running time. However, it is not clear if the paper is comparable with the state of the art and its presentation is not the best possible. Overall, I recommend weak-accept.


**Time Spent Reviewing:**

5

---

> ### Author Response · Authors · 2021-08-09
> **Comparison to other known algorithms; answers to questions for clarification**
>
> We appreciate the reviewer’s careful review and thoughtful comments.
>
> **How is the algorithm compared to the fastest known algorithm for weighted bipartite matching?**
>
> The Hungarian method remains one of the primary algorithms used in practice, despite the existence of many asymptotically better theoretical algorithms. As an added bonus, its implementations are publicly available.
>
> Another reviewer suggested running algorithms for min cost flow for comparison, which we agree would be interesting to test.  But based on the close connection between the Hungarian algorithm and flow-based methods, we believe that our approach (which is specialized for matchings) will still be significantly faster.
>
> Overall, from a purely theoretical perspective, as we discuss in the paper, our algorithm improves over the state of the art by approximately a $\log(nC)$ factor if the predictions are good enough.  An additional property of our approach is that we can use any maximum cardinality matching algorithm as a subroutine. Letting $T_{MCM}$ denote its running time, our algorithm will then run in $O(error * T_{MCM})$ time. Therefore, when the error is small, we essentially reduce the running time of minimum weight bipartite perfect matching to that of max cardinality matching.
> From a practical angle, since our algorithm runs the Hungarian method but initialized with learned duals, our method is extremely easy to implement.
>
>
> Thank you for the detailed comments, we will address them and improve the presentation. Below we answer the comments asking for clarification.
>
> **Theorem 3 does not provide the running time for learning $\hat y$.**
> We set $\hat y$ to be the median of the optimal duals of the sampled instances. Therefore, the running time depends on the running time for computing the min-cost perfect matching and the sample size. We will make this clear.
>
> **Proof of Theorem 19 (in appendix) is hard to follow. In Line 452, is it $2^d$ or $2^n$?**
> Thank you for finding this typo. It should be $2^n$.
>
> **Line 453: $y_i$ or $x^i$? Why is it clear that the number of hyperplane is $O(kn)$?**
> It should be $x^i$. We believe it was not clear because of this typo. Then, the number of hyperplanes that go through one of the k $x^i$ points and that are perpendicular to one of $e_1, e_2, …, e_n$ is clearly $O(kn)$.
>
> **Paragraph starting at line 346 in the experiment. The description is a bit unclear. What is $(i,j)$?**
> It means an edge with two end points $i$ and $j$; $i$ is on the left and $j$ on the right. After partitioning and clustering, we have $2k$ clusters of points, $k$ clusters on each side (left and right). We sample a point from each cluster at random and create an edge between two sampled points $i$ and $j$, where $i$ and $j$ belong to the left and right sides, respectively.

---

### Decision · Program_Chairs · 2021-09-27

**Decision:**

Accept (Oral)

**Comment:**

The paper presents an efficient learning-augmented algorithm for min cost matchings in bipartite graphs, by utilizing predictions of dual solutions. Both in theory and in experiments, the efficiency of the proposed algorithm (essentially the Hungarian algorithm with a "learned" warm-start) depends on the quality of predictions. When the predictions are "good", the algorithm is (significantly) more efficient than the baseline algorithm that does not use any predictions.

The reviewers appreciated several aspects of this paper, such as (i) the wide applicability of the Hungarian algorithm (ii) the generality of the proposed framework, (iii) the elegance of the solution and (iv)  the novelty of using predictions to improve the runtime as opposed to, say, the competitive ratio .  There were some concerns whether the Hungarian algorithm is an appropriate baseline, and whether the predictors are too simple, but they were deemed minor or were resolved during the rebuttal phase.